# Primary SARS-CoV-2 exposure by vaccination or infection shapes immune responses to omicron variants among a Spanish cohort

Otavio Ranzani [1,2,13] ✉, Carla Martín Pérez [1,13], Rocío Rubio [1,3], Anna Ramírez-Morros [4], Alfons Jimenez [1,5], Marta Vidal[1], Mar Canyelles [1,3], Cèlia Torres [1], Diana Barrios[1], Inocencia Cuamba [1,3,6], Pere Santamaria [7,8], Pau Serra[7], Luis Izquierdo [1,9], Josep Vidal-Alaball [4,10,11], Luis M. Molinos-Albert [1], Ruth Aguilar [1], Anna Ruiz-Comellas [4,10,11,12], Gemma Moncunill [1,9,14] & Carlota Dobaño [1,9,14] ✉

The comparison between vaccine-induced and infection-acquired adaptive immunity, and their co-occurrence —referred to as "hybrid immunity"— is of great interest and remains an area with significant knowledge gaps. Given that most of the population already has hybrid immunity to COVID-19, a key question is whether the order of infection-acquired and vaccine-induced immunity affects the immune response. Here, we analyze the humoral and T-cell responses in a Spanish cohort with longitudinal blood sampling spanning 2020-2023. We observe higher anti-RBD antibody levels against Omicron in individuals initially exposed to SARS-CoV-2 antigens via vaccination compared to those first exposed through natural infection. This difference diminishes with an increasing number of exposures. The dynamics of antibody levels over time correlate with clinical protection: those first-infected have higher protection early on, whereas those first-vaccinated show greater protection later, especially with the arrival of the Omicron variant. This phenomenon may reflect immune imprinting. In contrast to the humoral response, the T-cell response is higher in individuals first exposed through infection, although T-cell findings may be underpowered because of limited sample size. Our study provides valuable insights into the impact of initial antigen exposure on humoral and cellular responses to SARS-CoV-2.

The severe acute respiratory syndrome coronavirus 2 (SARS-CoV-2) has been a major challenge to society since December 2019. After the pandemic has eased, SARS-CoV-2 still appears as a main cause of morbidity and mortality, with ongoing waves driven by new variants[1,2].

Extensive research has been conducted on the immune response to SARS-CoV-2, including the role of the innate response, the antibodies targeting the trimeric spike (S) protein, and the cellular response mediated by T cells[3]. Among the antibodies targeting the S protein, those against the receptor-binding protein domain (RBD)

are crucial for neutralizing activity and protection against the disease[4–6].

The comparison between vaccine-induced and infection-acquired adaptive immunity, and their co-occurrence ("hybrid immunity"), is of great interest and remains an area with knowledge gaps[5,7,8]. Regarding antibody dynamics, a systematic review and meta-analysis showed higher peak antibody titers for infection-acquired antibodies compared to vaccine-induced antibodies, particularly in the first weeks post-exposure and after 6-months[9]. In contrast, some studies showed the opposite, with higher peaks of anti-RBD antibodies in first-vaccinated compared with first-infected individuals[10]. Additionally,

**Table 1 | Characteristics of cohort stratified by the first exposure being an infection or vaccination**

|  | First-infected (n = 197) | First-vaccinated (n = 160) | p-value[a] |
|---|---|---|---|
| Timepoint (%) |  |  | 0.813 |
| T10 | 28 (14.2) | 20 (12.5) |  |
| T10 and T11 | 159 (80.7) | 130 (81.2) |  |
| T11 | 10 (5.1) | 10 (6.2) |  |
| Age, mean (SD), years | 50.2 (11) | 47.1 (10) | 0.006 |
| Female, n (%) | 165 (83.8) | 138 (86.2) | 0.613 |
| Number of chronic comorbidities |  |  |  |
| mean (SD) | 0.78 (1.01) | 0.46 (0.73) |  |
| median [IQR] | 0 [0, 1] | 0 [0, 1] | 0.002 |
| Chronic respiratory disease, n (%)[b] | 8 (4.1) | 12 (7.5) | 0.24 |
| COPD, n (%) | 0 (0.0) | 3 (1.9) | 0.178 |
| Asthma, n (%) | 8 (4.1) | 10 (6.2) | 0.486 |
| Tobacco smoking status, n (%) |  |  | <0.001 |
| No | 137 (69.5) | 92 (57.5) |  |
| Previous smoker | 49 (24.9) | 38 (23.8) |  |
| Active smoker | 11 (5.6) | 30 (18.8) |  |
| Cardio-metabolic, n (%)[b] | 31 (15.7) | 11 (6.9) | 0.016 |
| Dyslipidemia, n (%) | 14 (7.1) | 4 (2.5) | 0.083 |
| Hypertension, n (%) | 13 (6.6) | 6 (3.8) | 0.339 |
| Diabetes, n (%) | 7 (3.6) | 0 (0.0) | 0.043 |
| Cardiovascular diseases, n (%) | 2 (1.0) | 2 (1.2) | >0.99 |
| Obesity, n (%) | 23 (11.7) | 9 (5.6) | 0.071 |
| Neurologic, n (%) | 2 (1.0) | 2 (1.2) | >0.99 |
| Gastrointestinal, n (%) | 9 (4.6) | 4 (2.5) | 0.451 |
| Chronic renal disease, n (%) | 0 (0.0) | 1 (0.6) | 0.917 |
| Immunosuppressed status, n (%)[b] | 20 (10.2) | 8 (5.0) | 0.109 |
| Autoimmune disease, n (%) | 13 (6.6) | 6 (3.8) | 0.339 |
| Cancer, n (%) | 6 (3.0) | 2 (1.2) | 0.435 |
| Other immunosuppression, n (%) | 2 (1.0) | 0 (0.0) | 0.572 |
| Hypothyroidism, n (%) | 17 (8.6) | 10 (6.2) | 0.519 |
| Depression, n (%) | 5 (2.5) | 2 (1.2) | 0.625 |
| Pregnancy, n (%) | 3 (1.5) | 1 (0.6) | 0.767 |
| History of allergy, n (%) | 33 (16.8) | 13 (8.1) | 0.024 |

[a]The p-values were estimated from Chi-square tests with Yates's correction for categorical variables, from independent T test for age and from Mann–Whitney test for number of chronic comorbidities. All tests were two-sided.
[b]The individuals can have more than one comorbidity within the category, thus the combined variable does not necessarily sum to each comorbidity. COPD represents chronic obstructive pulmonary disease.

the cellular and humoral responses after a vaccine shot appear to be broader and more sustained in those individuals with a previous infection compared to those with only a previous vaccine dose[5,11]. Nevertheless, the difference in the humoral response seems to disappear over time since the prime-exposure (i.e., via vaccine or infection)[9] and with additional exposures, such as after a third exposure to SARS-CoV-2 antigens[7,12,13]. With the advent of the Omicron variant and its descendants, the viral escape from neutralizing and binding antibody titers was significant[12,14], increasing the number of breakthrough infections and raising new questions about the differences between vaccine-induced and infection-acquired immune responses[7,12,15].

Given that most of the population already has hybrid immunity to COVID-19, one knowledge gap is whether the order of infection-acquired and vaccine-induced immunity affects the resulting immune response. Indeed, infection-acquired immunity to SARS-CoV-2 exhibits great variability[12], including low-responder profiles, failure to mount long-lasting immunity, and delayed response to vaccination in a subset of individuals[16–18]. These effects are likely to occur during the first-ever exposure to the SARS-CoV-2 antigens. Whether they impact long-term and broad immunity in the scenario of multiple previous vaccine doses and infections, where Omicron is the predominant variant, is unknown.

We leveraged a well-characterized longitudinal cohort of health-care workers (CovidCatCentral)[19,20] to evaluate whether a first exposure to SARS-CoV-2 antigens via infection or vaccination is associated with a better anti-RBD antibody response to Omicron variants in individuals with hybrid immunity. Subsequently, we explored whether the dynamics of the anti-RBD antibody response to the ancestral Wuhan strain differed by first exposure groups, as well as their T-cell response and clinical protection.

## Results

### Population characteristics and history of exposures

The population characteristics are shown in Table 1. The majority of individuals had samples available from both study timepoints, T10 and T11, accounting for 81% of all samples. Individuals who were first-infected were older, had more cardio-metabolic comorbidities, a higher prevalence of allergy, and were less frequently active smokers compared with the first-vaccinated group.

The temporal distribution of the number of previous asymptomatic and symptomatic infections and of vaccine doses is shown in Fig. 1, Table 2 and Supplementary eTable 1. Overall, the number of previous exposures was higher in the first-infected group compared with the first-vaccinated group. The number of infections was higher in the first-infected group when considering any infection, mainly driven by non-Omicron infections (i.e., Wuhan). The number of asymptomatic infections was comparable between groups in terms of number of exposures and for non-Omicron and Omicron periods (Supplementary eTable 1). The largest difference among Omicron variants concerned BQ.1, being more frequent on first-vaccinated group (Supplementary eTable 1). Overall, the number of previous vaccinations was higher in the first-vaccinated group. The most administered vaccine type was Pfizer (n = 840, 83%), followed by Moderna (n = 162, 16%), Astrazeneca (n = 10, 1%) and Janssen (n = 1, 0.1%). Regarding the first vaccine exposure, Pfizer was 100% for those first-vaccinated and 94% for those first-infected. The number of bivalent vaccines was similar between groups. The distribution of the history of previous infections and vaccine types are shown in Fig. 2. The time from the last exposure (i.e., last vaccine or infection) relative to the blood sampling used in the analysis was 173 [27–299] days for T10 and 224 [164–380] days for T11; and it was shorter for the first-vaccinated group compared to the first-infected group for T10 and similar for T11 (Table 2).

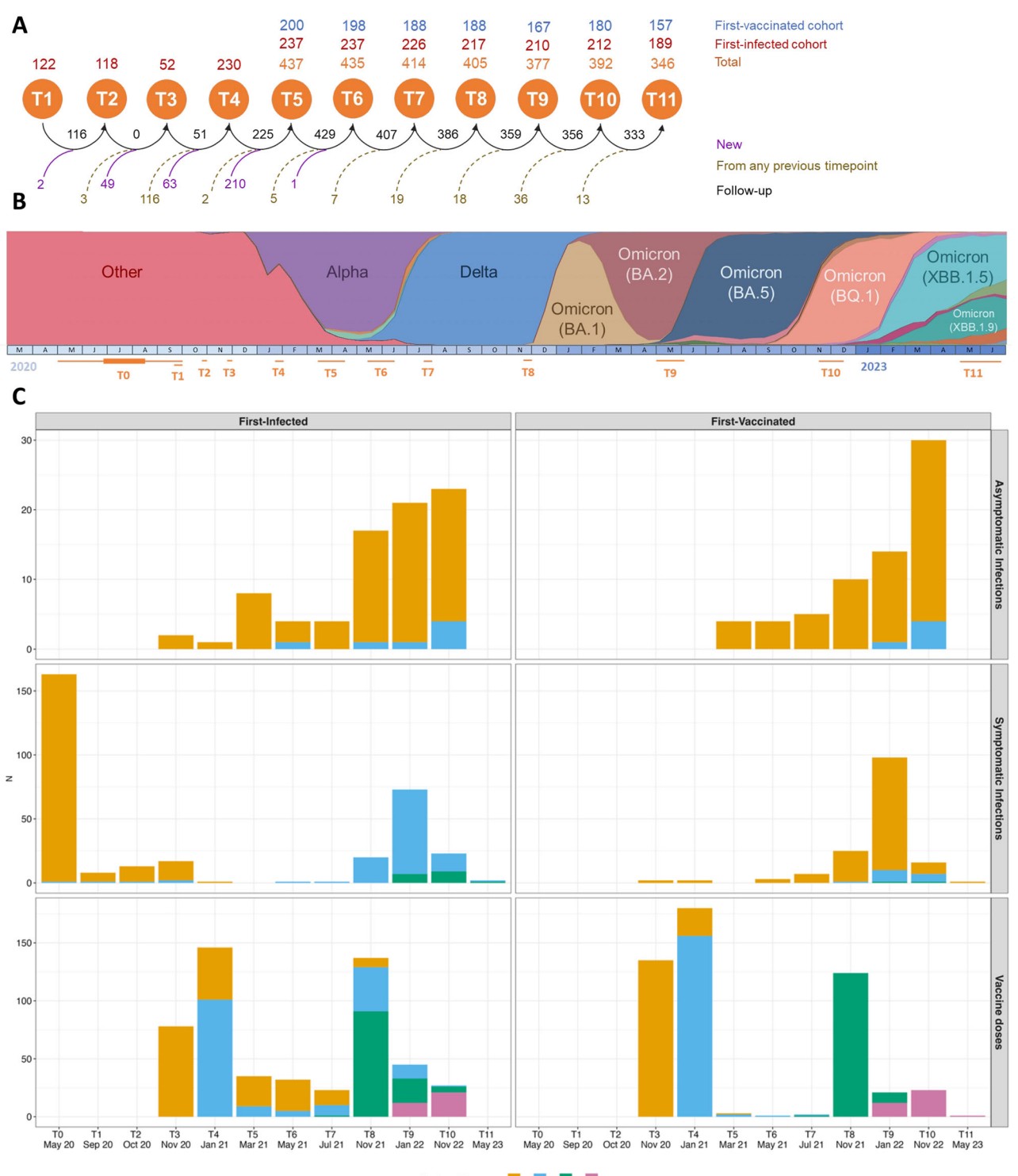

**Fig. 1 | CovidCatCentral Cohort history. A** Number of individuals in each of the CovidCatCentral cohort timepoints for the first-vaccinated and first-infected groups before study inclusion selection. The blue numbers represent the first-vaccinated, red numbers represent the first-infected, and the orange numbers show the total number of individuals at each time point. Magenta numbers indicate new individuals entering the study, darkgold numbers indicate individuals from any previous timepoints but not consecutive, and black numbers indicate individuals who followed up from one consecutive timepoint to the next. **B** Prevalence of the different COVID-19 variants circulating in Spain from 2020 to 2023 and related to the timepoints. **C** Temporal distribution of infections and vaccinations stratified by first exposure type among the analyzed population. Omicron infections started between T8 and T9. T: timepoint. Source data are provided in the source data.

## Anti-RBD antibody correlations

We measured IgA and IgG in 646 plasma samples (337 from T10 and 309 from T11). The correlations between anti-RBD IgA and IgG levels to Wuhan and six different Omicron lineages for each timepoint are shown in Supplementary eFig. 1. The correlations between anti-RBD Wuhan and anti-RBD Delta responses were higher than between anti-RBD Wuhan and the other anti-RBDs targeting Omicron lineages. The majority of correlation coefficient values < 0.80 occurred when contrasting anti-RBD XBB with others. Overall, correlations between IgA and IgG were <0.40 across all timepoints and RBDs. The correlations of

**Table 2 | Infections and Vaccination history before T10 and T11 timepoints**

|  | T10 | | | T11 | | |
|---|---|---|---|---|---|---|
|  | First-infected (n = 197) | First-vaccinated (n = 160) | p-value | First-infected (n = 197) | First-vaccinated (n = 160) | p-value[a] |
| Number of previous exposures,[b] mean (SD) | 4.50 (1.00) | 4.31 (0.66) | 0.034 | 4.70 (1.06) | 4.44 (0.77) | 0.012 |
| Number of previous exposures,[b] n (%) |  |  | <0.001 |  |  | 0.001 |
| 3 | 33 (16.8) | 13 (8.1) |  | 25 (12.7) | 12 (7.5) |  |
| 4 | 68 (34.5) | 90 (56.2) |  | 63 (32.0) | 80 (50.0) |  |
| 5 | 64 (32.5) | 52 (32.5) |  | 66 (33.5) | 53 (33.1) |  |
| 6 | 29 (14.7) | 5 (3.1) |  | 34 (17.3) | 15 (9.4) |  |
| 7 | 2 (1.0) | 0 (0.0) |  | 8 (4.1) | 0 (0.0) |  |
| 8 | 1 (0.5) | 0 (0.0) |  | 1 (0.5) | 0 (0.0) |  |
| Infections |  |  |  |  |  |  |
| Number of previous infections,[c] mean (SD) | 1.95 (0.69) | 1.33 (0.52) | <0.001 | 2.04 (0.71) | 1.38 (0.56) | <0.001 |
| Number of previous infections,[c] n (%) |  |  | <0.001 |  |  | <0.001 |
| 1 | 49 (24.9) | 111 (69.4) |  | 41 (20.8) | 105 (65.6) |  |
| 2 | 110 (55.8) | 45 (28.1) |  | 111 (56.3) | 49 (30.6) |  |
| 3 | 36 (18.3) | 4 (2.5) |  | 41 (20.8) | 6 (3.8) |  |
| 4 | 2 (1.0) | 0 (0.0) |  | 4 (2.0) | 0 (0.0) |  |
| Type of first infection,[c] n (%) |  |  | <0.001 |  |  | <0.001 |
| Wuhan | 193 (98.0) | 0 (0.0) |  | 193 (98.0) | 0 (0.0) |  |
| Alpha | 4 (2.0) | 13 (8.1) |  | 4 (2.0) | 13 (8.1) |  |
| Delta | 0 (0.0) | 26 (16.2) |  | 0 (0.0) | 26 (16.2) |  |
| Omicron BA.1 | 0 (0.0) | 42 (26.2) |  | 0 (0.0) | 42 (26.2) |  |
| Omicron BA.2 | 0 (0.0) | 27 (16.9) |  | 0 (0.0) | 27 (16.9) |  |
| Omicron BA.4/5 | 0 (0.0) | 32 (20.0) |  | 0 (0.0) | 32 (20.0) |  |
| Omicron BQ.1 | 0 (0.0) | 19 (11.9) |  | 0 (0.0) | 19 (11.9) |  |
| Omicron XBB.1 | 0 (0.0) | 1 (0.6) |  | 0 (0.0) | 1 (0.6) |  |
| Number of previous Wuhan infections,[c] n (%) |  |  | <0.001 |  |  | <0.001 |
| 0 | 4 (2.0) | 160 (100.0) |  | 4 (2.0) | 160 (100.0) |  |
| 1 | 187 (94.9) | 0 (0.0) |  | 187 (94.9) | 0 (0.0) |  |
| 2 | 6 (3.0) | 0 (0.0) |  | 6 (3.0) | 0 (0.0) |  |
| Number of previous Alpha infections,[c] n (%) |  |  | 0.665 |  |  | 0.665 |
| 0 | 180 (91.4) | 147 (91.9) |  | 180 (91.4) | 147 (91.9) |  |
| 1 | 16 (8.1) | 13 (8.1) |  | 16 (8.1) | 13 (8.1) |  |
| 2 | 1 (0.5) | 0 (0.0) |  | 1 (0.5) | 0 (0.0) |  |
| Number of previous Delta infections,[c] n (%) |  |  | 0.12 |  |  | 0.12 |
| 0 | 175 (88.8) | 132 (82.5) |  | 175 (88.8) | 132 (82.5) |  |
| 1 | 21 (10.7) | 28 (17.5) |  | 21 (10.7) | 28 (17.5) |  |
| 2 | 1 (0.5) | 0 (0.0) |  | 1 (0.5) | 0 (0.0) |  |
| Number of previous Omicrons infections,[c] n (%) |  |  | <0.001 |  |  | <0.001 |
| 0 | 69 (35.0) | 16 (10.0) |  | 58 (29.4) | 15 (9.4) |  |
| 1 | 111 (56.3) | 116 (72.5) |  | 116 (58.9) | 110 (68.8) |  |
| 2 | 17 (8.6) | 28 (17.5) |  | 23 (11.7) | 35 (21.9) |  |
| Vaccines |  |  |  |  |  |  |
| Number of previous vaccines, mean (SD) | 2.55 (0.82) | 2.98 (0.55) | <0.001 | 2.65 (0.89) | 3.06 (0.62) | <0.001 |
| Number of previous vaccines, n (%) |  |  | <0.001 |  |  | <0.001 |
| 1 | 23 (11.7) | 0 (0.0) |  | 22 (11.2) | 0 (0.0) |  |
| 2 | 61 (31.0) | 26 (16.2) |  | 57 (28.9) | 26 (16.2) |  |
| 3 | 95 (48.2) | 112 (70.0) |  | 85 (43.1) | 98 (61.3) |  |
| 4 | 18 (9.1) | 22 (13.8) |  | 33 (16.8) | 36 (22.5) |  |
| Type of first vaccine |  |  | 0.01 |  |  | 0.01 |

**Table 2 (continued) | Infections and Vaccination history before T10 and T11 timepoints**

| | T10 | | | T11 | | |
|---|---|---|---|---|---|---|
| | First-infected (n = 197) | First-vaccinated (n = 160) | p-value | First-infected (n = 197) | First-vaccinated (n = 160) | p-value[a] |
| Pfizer (BNT162b2) | 186 (94.4) | 160 (100.0) | | 186 (94.4) | 160 (100.0) | |
| Moderna (mRNA-1273) | 5 (2.5) | 0 (0.0) | | 5 (2.5) | 0 (0.0) | |
| Astrazeneca (ChAdOx1) | 6 (3.0) | 0 (0.0) | | 6 (3.0) | 0 (0.0) | |
| Number of previous Pfizer | | | <0.001 | | | <0.001 |
| 0 | 9 (4.6) | 0 (0.0) | | 6 (3.0) | 0 (0.0) | |
| 1 | 39 (19.8) | 0 (0.0) | | 38 (19.3) | 0 (0.0) | |
| 2 | 91 (46.2) | 86 (53.8) | | 90 (45.7) | 79 (49.4) | |
| 3 | 48 (24.4) | 65 (40.6) | | 46 (23.4) | 66 (41.2) | |
| 4 | 10 (5.1) | 9 (5.6) | | 17 (8.6) | 15 (9.4) | |
| Number of previous Moderna | | | 0.608 | | | 0.602 |
| 0 | 111 (56.3) | 87 (54.4) | | 110 (55.8) | 86 (53.8) | |
| 1 | 85 (43.1) | 73 (45.6) | | 86 (43.7) | 74 (46.2) | |
| 2 | 1 (0.5) | 0 (0.0) | | 1 (0.5) | 0 (0.0) | |
| Number of previous Adeno-virus vaccines[±] | | | 0.055 | | | 0.036 |
| 0 | 190 (96.4) | 160 (100.0) | | 189 (95.9) | 160 (100.0) | |
| 1 | 4 (2.0) | 0 (0.0) | | 5 (2.5) | 0 (0.0) | |
| 2 | 3 (1.5) | 0 (0.0) | | 3 (1.5) | 0 (0.0) | |
| Previous bivalent vaccine, n (%) | 1 (0.5) | 3 (1.9) | 0.475 | 17 (8.6) | 13 (8.1) | >0.99 |
| Times | | | | | | |
| Days from closest exposure, median [IQR] | 189 [44, 319] | 153 [21, 224] | 0.005 | 227 [165, 379] | 211 [162, 382] | 0.514 |
| Days from closest infection, median [IQR] | 222 [140, 695] | 184 [110, 295] | <0.001 | 357 [183, 548] | 326 [173, 404] | <0.001 |
| Days from closest vaccine dose, median [IQR] | 340 [303, 356] | 347 [331, 360] | 0.114 | 504 [293, 533] | 514 [339, 530] | 0.437 |

[a]The p-values were estimated from Chi-square tests with Yates's correction for categorical variables, from independent T tests for all continuous variables except for the three times, which were compared with Mann–Whitney tests. All tests were two-sided.
[b]Includes vaccine doses, symptomatic and asymptomatic infections.
[c]Includes symptomatic and asymptomatic infections. ± Includes one Janssen and other are all Astrazeneca.

anti-RBDs antibody levels between T10 and T11 were positive and close to 0.70.

## Association between antibody titers at T10 and T11 and type of first exposure

The unadjusted average MFI values, accounting for the number of previous exposures, for both IgA and IgG across the seven RBDs were higher for those first-vaccinated than first-infected (35%, 95% CI, 20 to 51, for IgA and 53%, 95% CI, 33 to 76, for IgG against RBD XBB), although this difference was less clear for IgA and anti-RBD Wuhan and anti-RBD Delta (21%, 95% CI, −2 to 48, for IgA and 54%, 95% CI, 35 to 76, for IgG against RBD Wuhan, Fig. 3-M0 and Supplementary eFig. 2). When adjusting for the sequential set of potential confounders, this difference remained showing greater MFI values for first-vaccinated than first-infected (35%, 95% CI, 15 to 58, for IgA and 70%, 95% CI, 42 to 105, for IgG against XBB, Fig. 3, main model, M5). For example, in the four sensitivity analyses, the pattern of association remained similar.

In the main model (M5), the covariates associated with an increased IgG response were previous number of Omicron infections for some RBDs and, remarkably, previous number of non-bivalent vaccines (e.g., three previous for non-bivalent vaccines compared with one previous non-bivalent for anti RBD XBB: 44%, 95% CI, 14 to 82, Supplementary eTable 2) and having received at least one bivalent vaccine for the majority of RBDs (e.g., anti-RBD XBB, 64%, 95% CI, 40 to 91). For IgA response, the covariates did not have a clear pattern associated with increased antibody response, except for the bivalent

vaccine (anti-RBD BA4.5, 25%, 95% CI, 7 to 46, Supplementary eTable 2).

Overall, the results analyzing the anti-RBD/anti-S ratio showed similar pattern to the main analysis, although uncertainty was higher for the IgA/anti-S ratio and anti-RBD XBB.1/anti-S ratio (Supplementary eFig. 3).

We further explored whether the observed differences between first-vaccinated and first-infected groups varied according to the number of previous exposures (infections or vaccinations). We observed that the larger antibody response in first-vaccinated individuals was mainly driven by those with up to three previous exposures and that this difference decreased as the individuals had been exposed to more exposures (Fig. 4). This pattern was almost linear and consistently observed for IgA across all RBDs and for IgG anti-RBD Wuhan and anti-RBD Delta. When evaluating IgG against Omicron lineages, the pattern of decreasing the difference between first-vaccinated and first-infected was present, and seems to have a threshold, being the interaction mainly driven by the difference occurring before and after four exposures. There was no evidence of this pattern regarding the first-exposure group for IgG against anti-RBD XBB, which remained almost flat across the number of exposures.

We compared the differences between first-vaccinated and first-infected individuals considering their actual previous history and sequence of exposures (Fig. 5). The histories with consistent increased IgG response against RBD for the first-vaccinated compared with first-infected groups were the sequence of two vaccine doses followed by

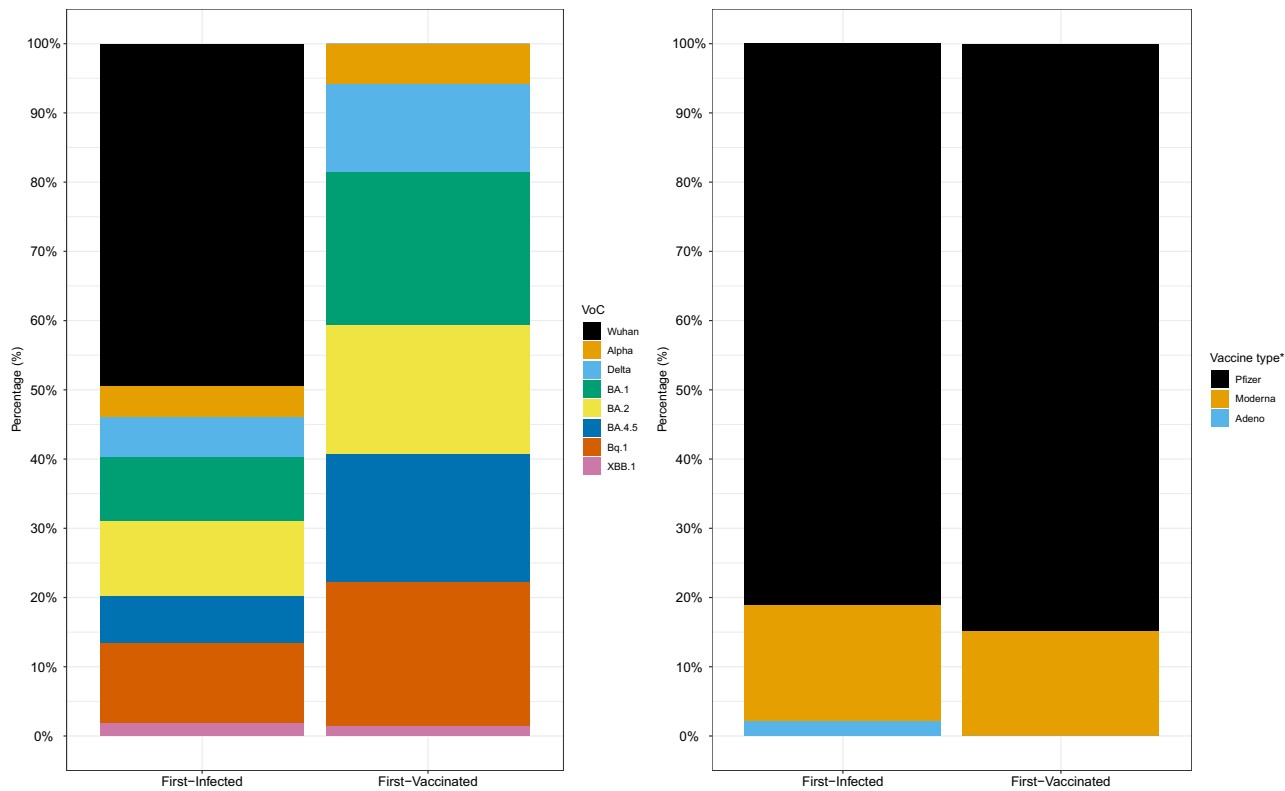

**Fig. 2 | The entire history of previous infections and previous vaccines stratified by first-infected and first-vaccinated groups.** * There was only 1 Janssen that we grouped with Astrazeneca as Adenovirus-based vaccines. Source data are provided in the source data.

an infection compared with an initial infection followed by one vaccination and one infection (VacVacInf vs InfVacInf), and the history of three vaccine doses followed by an infection (VacVacVacInf vs InfVacVacInf). The history VacVacVacInf vs. InfVacVacInf showed increased response for IgA against Omicron RBDs. When we explored FDR for multiplicity correction, we did not observe any major changes in the interpretation of the results analyzing *p*-values < 0.05, except for three comparisons where Bonferroni was close to *p* = 0.05 and FDR was below 0.05 (two contrasts for IgA and anti-RBD Delta and one contrast for anti-RBD BQ1.1, Fig. 5 and Supplementary eFig. 4).

### Kinetics of antibodies against N, RBD and S Wuhan since March 2021 (T5) by first exposure

When we looked at the temporal evolution of IgA and IgG to N, S and RBD from its first assessments in 2021, we observed that anti-N antibody levels, for both IgA and IgG, were overall lower in first-vaccinated than in first-infected individuals across all timepoints, with larger difference after T9. Regarding anti-S and anti-RBD levels, both IgA and IgG were lower for the first-vaccinated individuals during the first timepoints (T5 to T8), and then reversed after the T9 (Fig. 6). Sensitivity analyses regarding alternative parameterizations of LOWESS and the use of GAM to model the longitudinal evaluation showed similar patterns (Supplementary eFig. 5).

### Association between magnitude of T-cell responses at T11 and type of first exposure

We analyzed the magnitude of T-cell responses in a subset of 49 individuals (21 from the first-infected and 28 from the first-vaccinated groups) at T11. All tested individuals had detectable T-cell responses against S or N + M from Wuhan. S-specific T-cell responses were more robust than those to N + M (Supplementary eFig. 6A) although they were correlated (Spearman rho range 0.69-0.87, p < 0.05, Supplementary eFig. 6B). Most S-specific T cells secreted IFN-γ, while most

N + M-specific T cells secreted IL-2. Within S-specific T cells, the first-vaccinated group had higher proportion of IL-2 secreting T cells (37.8%) compared to the first-infected group (21.8%) (Supplementary eFig. 6A). In contrast to the pattern observed for antibodies, the first-infected group showed significantly increased magnitude of T-cell responses compared to the first-vaccinated group three years later to both S (3.5 and 1.9 times higher for IFN-γ and IFN-γ + IL-2, respectively) and N + M (3.2, 2.7 and 2.7 times higher for IFN-γ, IL-2 and IFN-γ + IL-2, respectively) (Fig. 7A). In the first-vaccinated group no correlations were found between the magnitude of T-cell responses and antibody levels, except for S-specific IFN-γ secreting T cells with IgG levels anti-RBD from Delta variant (Spearman rho 0.44, <0.05) (Fig. 7B). In contrast, within the first-infected group, the magnitude of T-cell responses to N + M was moderately correlated with IgG levels against Wuhan and other variants (Spearman rho range: 0.42-0.66, p < 0.05), while a negative correlation was observed between T-cell responses and IgA levels, particularly against RBD from Omicron variants (Spearman rho ranges −0.46 to −0.58, <0.05, Fig. 7B).

### Association between protection against breakthrough infection and first exposure

Having an initial vaccination as opposed to being first-infected was associated with an increased risk of symptomatic and asymptomatic breakthrough infections during the T8–T9 period (HR 3.48; 95% CI, 1.05–11.5). However, there was a shift in this risk for T9-T10 (HR 0.50; 95% CI, 0.31-0.82) and T10-T11 (HR 0.71, 95% CI, 0.43 to 1.17) periods, when being first-vaccinated was associated with a decreased risk of infection (Supplementary eTable 3).

## Discussion

In a longitudinal evaluation of 357 individuals with hybrid immunity to SARS-CoV-2, the first exposure to SARS-CoV-2 antigens via a natural infection resulted in lower IgA and IgG levels against Omicron variant

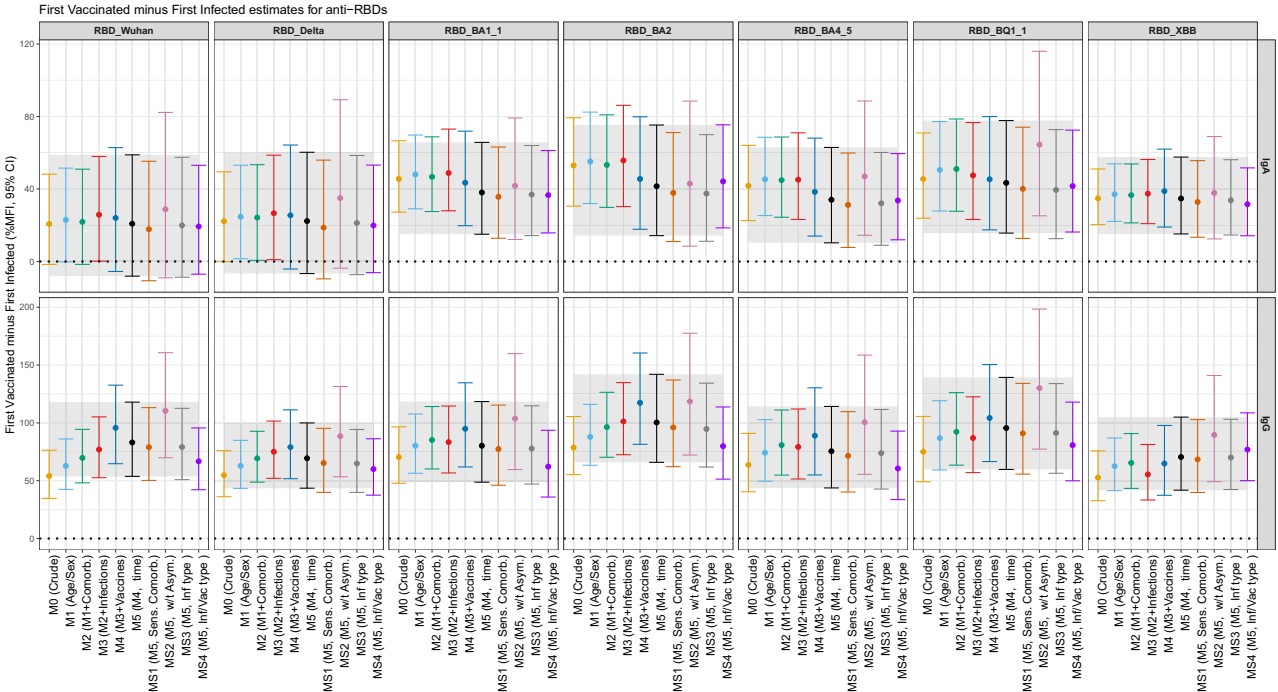

**Fig. 3 | Percent change on IgA and IgG antibody responses to RBD at T10 and T11 in first-vaccinated group *minus* first-infected group\*.** \*Estimates from linear mixed models estimating the %MFI increase in those first-vaccinated compared to the first-infected group (reference). The models accounted for repeated measurements in the same individual with a random intercept per individual. The models were adjusted as follows: M0, crude; M1, M0 + age (restricted cubic spline with 3 df) and sex; M2, M1 + number of chronic comorbidities and tobacco smoking status; M3, M2 + number of non-Omicron and Omicron symptomatic infections (as factors), and number of non-Omicron and Omicron asymptomatic infections (as factors); M4, M3 + number of non-bivalent and bivalent vaccines (as factors); and finally, M5 (main model), M4 + days from last infection (restricted cubic spline with 3 df) and days from last vaccine (restricted cubic spline with 3 df). MS1, M5 but

expanding the number of chronic comorbidities as three binary factors (cardio-metabolic, immunosuppressed and previous allergy); MS2, running M5 in those individuals without any history of asymptomatic infections; MS3, M5 but instead of non-bivalent/valent, using number of mRNA and adenovirus-based vaccines; MS4, MS3 but instead of number of non-Omicron/Omicron, using number of previous infections by each variant of concern. For each model, there were 337 samples from T10 and 309 samples from T11, totaling 646 samples from 357 individuals. The central circles represent the point estimates, and the vertical bars indicate the 95% confidence intervals. Source data are provided in the source data. df: degrees of freedom. MFI: Median fluorescence intensity. RBD: Receptor binding domain. T: timepoint. M: model.

RBDs compared with those first exposed via vaccination after an average of 420 days since their first exposure. This difference appeared to decrease as the total number of previous exposures increased. When evaluating a long series of antibody levels against Wuhan antigens, first-vaccinated individuals had lower levels of anti-RBD antibodies and anti-S antibodies at the beginning of follow-up, as previously described[9,21], which shifted after the appearance of Omicron. In contrast, T-cell responses against Wuhan antigens were of greater magnitude in those first-infected compared to those first-vaccinated; this conclusion is limited by the relatively small cellular assay sample size.

The importance of the first exposure to SARS-CoV-2 antigens was studied early in the pandemic[9,21]. However, few studies have evaluated its impact in a scenario mimicking the current situation, characterized by hybrid immunity and the circulation of the Omicron variant. Srivastava et al.[7], in a cohort of 496 individuals in New York City, observed similar antibody dynamics between groups. They reported higher anti-S levels in the first-infected group up to the third vaccine dose, after which the levels became comparable. Contrasting with our results, Srivastava *et al.* found the first-infected vs first-vaccinated difference disappeared after the third dose, while we observed this difference shifted. This discrepancy may be due to different antibody types and assays, in addition to differences in the adjustment for potential confounders. Importantly, the difference in anti-RBD antibody levels decreased with increasing exposures, except for anti-RBD XBB. To

the best of our knowledge, this is the first study showing this phenomenon.

Regarding protection against breakthrough infections, we observed a parallelism between anti-RBD antibodies and clinical protection, i.e., when first-infected individuals had higher anti-RBD antibody levels, they also had less risk of breakthrough infections compared to the first-vaccinated group, and this shifted when first-vaccinated group had higher anti-RBD antibody levels. Srivastava *et al.* found a similar pattern regarding this protection against breakthrough infections; nevertheless, they observed similar anti-S levels[7]. This indirectly reinforces the role of anti-RBD antibodies as being closer to neutralizing functions than anti-S.

In contrast with the previous findings regarding humoral and clinical protection assessment, the T-cell responses against Wuhan were more robust in those first-infected compared with the first-vaccinated at T11, even though all individuals had hybrid immunity before T10 and T11. The literature shows that T-cell response is a marker for protection against progression to severe disease, a fact that explains the protection observed against severe disease even when the neutralizing antibody levels against Omicron are relatively low[22]. Discordance between T-cell response and neutralizing antibodies has also been described, particularly in mild infections[23]. More robust cellular responses in those first-infected compared with first-vaccinated have been reported[24–27], such as a durable transcriptional and epigenetic signature of inflammation in S-specific memory CD4+ T cells after two years of the first exposure[26]. The first contact via infection exposes the

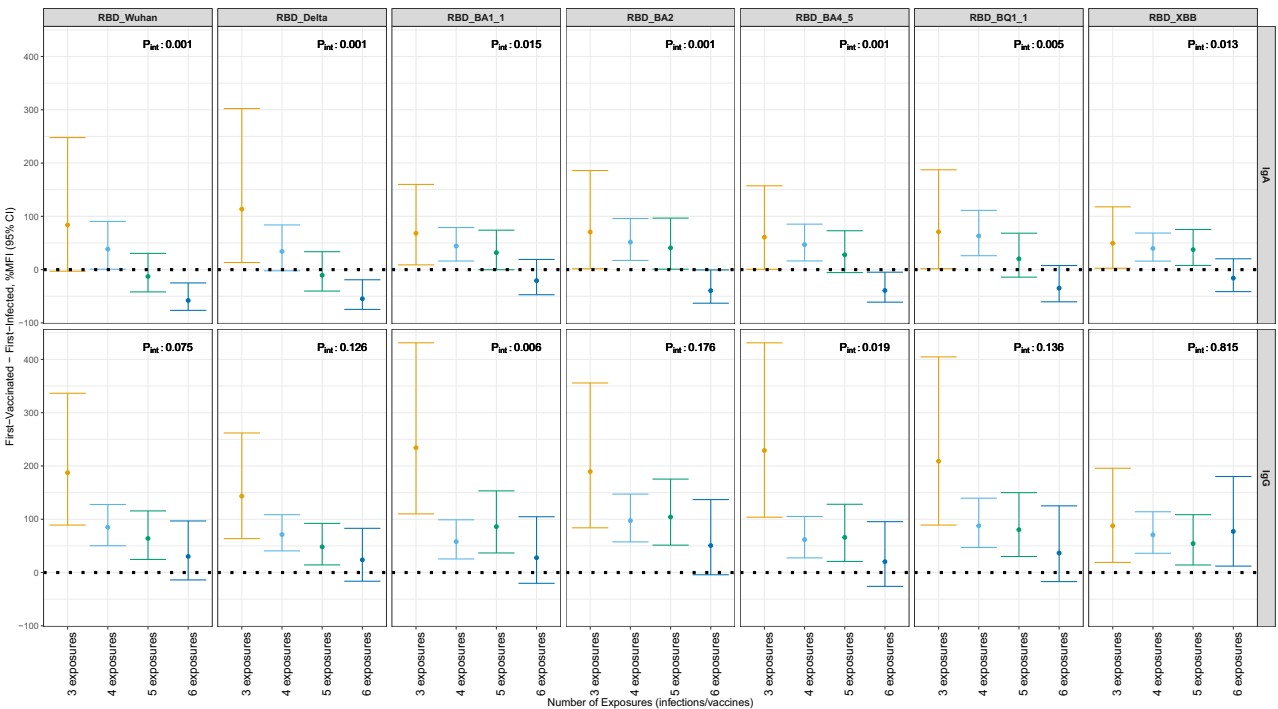

**Fig. 4 | Interaction between number of previous exposures and the percent change in IgA and IgG against RBD contrasting the first-vaccinated group minus the first-infected group at T10 and T11\*.** \* Estimates from linear mixed models estimating the %MFI increase in those first-vaccinated compared to the first-infected group (reference). The models accounted for repeated measurements in the same individual with a random intercept per individual. The model was adjusted as the main model (M5): age (restricted cubic spline with 3 df) + sex + number of chronic comorbidities and tobacco smoking status + number of non-Omicron and Omicron symptomatic infections (as factors) + number of non-Omicron and Omicron asymptomatic infections (as factors) + number of non-

bivalent and bivalent vaccines (as factors) + days from last infection (restricted cubic spline with 3 df) + days from last vaccine (restricted cubic spline with 3 df), with an interaction term between the main exposure and number of previous exposures (as a factor). For each isotype and variant of concern model, there were 334 samples from T10 and 306 samples from T11, totaling 640 samples from 354 individuals. The central circles represent the point estimates, and the vertical bars indicate the 95% confidence intervals. $P_{int}$ represents the two-sided *p*-value from the likelihood ration test for the interaction term. Source data are provided in the source data. df: degrees of freedom. MFI: median fluorescence intensity. M: model. T: timepoint. RBD: Receptor binding domain.

immune system to a wider repertoire of antigens, including mucosal antigen contact, exposure to different viral loads and a broader involvement of the innate immune response, likely providing a broader and more sustained T-cell response. Interestingly, we observed no correlation between T-cell responses and IgG levels among those first-vaccinated, despite they having higher anti-RBD antibody levels; by contrast, there were consistent positive correlations between T-cell responses and IgG levels among those in the first-infected group.

Some studies provided some insights on the potential mechanisms of the differences between first-infected and first-vaccinated, beyond the difference in T-cell response magnitude. Tejedor Vaquero et al. observed differences between previously infected and non-infected individuals in the recall of IgG subclasses to RBD after mRNA vaccination[28]. After the first dose, those previously infected had higher levels of anti-RBD IgG for all subclasses, except IgG3 for Wuhan and Beta, Gamma and Delta variants. In the second dose, those previously infected had a less robust response, and those without previous infection showed an increased recall for IgG1 against the variants of concerns. Pérez-Alós *et al.* also showed that primary infection before Omicron was essential for a robust IgA response, which might also explain why the difference between first-infected and first-vaccinated is less clear for IgA in our results[27]. Importantly, because of Spain's vaccine rollout and epidemic waves, individuals in the first-infected group were primarily infected in 2020–2021, during waves dominated by pre-Omicron variants (predominantly with Wuhan, but four individuals with Alpha). In contrast, individuals in the first-vaccinated group

were largely infected post-vaccination, when Omicron and its sub-variants predominated. Given the marked antigenic differences between Omicron and earlier variants, the first infecting strain likely acts as an effect modifier in the effect of first-exposure type on antibody levels. This contrast is an inherent aspect of our research question and contributes to the observed differences.

We observed a main change in the antibody dynamics after the emergence of Omicron. Reynolds et al. evaluated a cohort of health-care workers with three COVID-19 vaccines and different histories of infections to SARS-CoV-2 variants before Omicron[29]. Evaluating different aspects of humoral and cellular immune response, they observed there was a boost response upon an Omicron infection, but this boosting was much less robust in those with a prior Wuhan infection, a phenomenon defined by them as "hybrid immune damping". This could explain our main findings of a greater anti-RBD antibody response in those first-vaccinated compared with those first-infected (all with Wuhan). This imprinting for Omicron responses was also observed in other cohorts[27]. Interestingly, this imprinting wanes as the number of repeated Omicron exposures increases[30], which might also reflect our findings between first-infected vs first-vaccinated when considering the number of exposures.

Our study has some strengths. We report a well-characterized cohort of healthcare workers with close surveillance and repeated blood sampling. We also considered symptomatic and asymptomatic infections, made possible by repeated serological multiplex evaluation, which is not always possible in cohort studies evaluating humoral and cellular immune response, and clinical protection. These factors

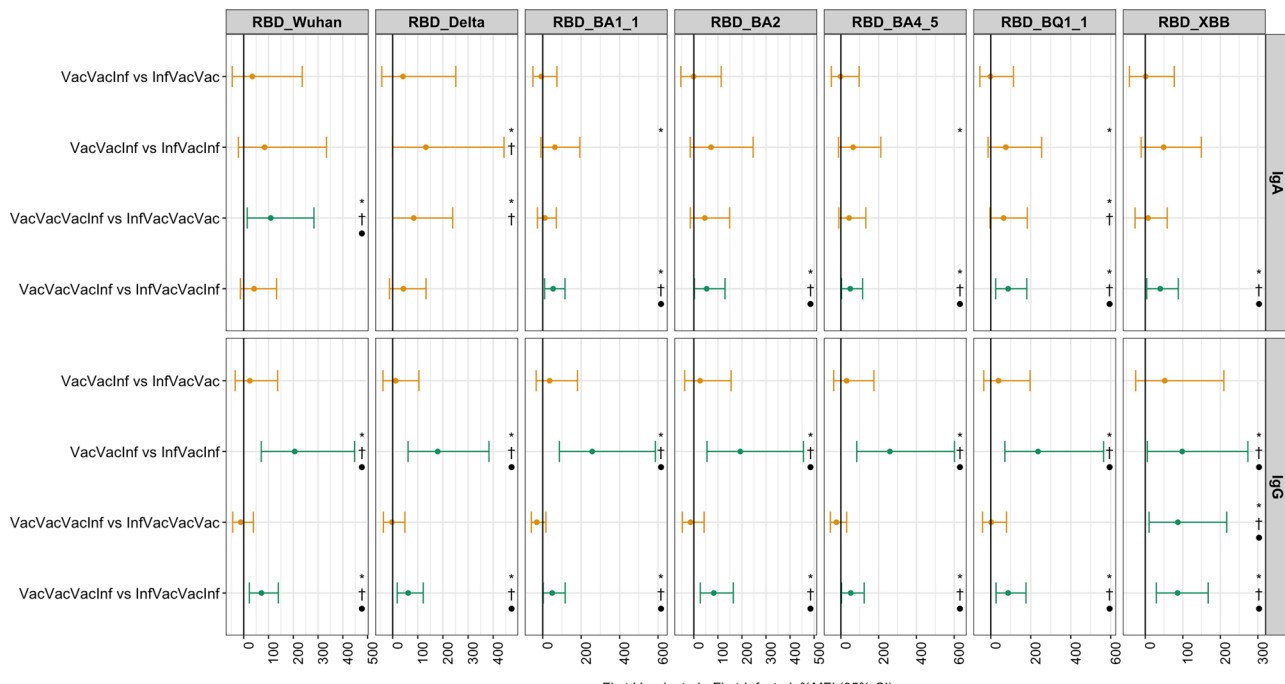

**Fig. 5 | Percent change contrasting first-vaccinated group minus first-infected group in four pairs of sequences of previous exposures for IgA and IgG against RBD at T10 and T11\*.** \* Estimates from linear mixed models estimating the %MFI increase in different exposure histories. The models accounted for repeated measurements in the same individual with a random intercept per individual. The model was adjusted adapting the main model (M5), i.e., excluding the number of previous infections and vaccines and using the history per se: age (restricted cubic spline with 3 df) + sex + number of chronic comorbidities and tobacco smoking status + days from last infection (restricted cubic spline with 3 df) + days from last vaccine (restricted cubic spline with 3 df) and a factor with the history of the most common combinations. For each model, there were 337 samples from T10 and 309 samples from T11, totaling 646 samples from 357 individuals. The central circles represent the point estimates, and the vertical bars indicate the 95% confidence intervals corrected by multiple comparisons with the Bonferroni method. Green color represents confidence intervals that do not include zero. The symbols represent the contrasts that achieved the statistical threshold of $p \leq 0.05$ accordingly with multiplicity correction methods: \*$p \leq 0.05$ without any correction; †$p \leq 0.05$ with false-discovery rate (FDR) correction, and ●$p \leq 0.05$ with Bonferroni correction. All $p$-values analysed in Fig. 5 can be seen at Supplementary eFig. 4. Source data are provided in the source data. df: degrees of freedom. MFI: median fluorescence intensity. M: model. T: timepoint. RBD: Receptor binding domain.

allowed us to adjust the estimates by the full history of previous SARS-CoV-2 antigen exposure in the cohort. Finally, we evaluated five Omicron antigens, which gives a broad evaluation of humoral response.

We also have several limitations. First, we did not evaluate the mucosal response, which has a significant role on protection and might differ between first-infected and first-vaccinated groups[12]. Second, we could not evaluate more exposure histories than the four reported, nor detailed differences by vaccine type because of sample size. However, we do not expect any major differences by type of vaccine since virtually 100% of the first vaccine in both groups were mRNA, and because estimates remained stable when adjusting by the number of previous mRNA and adenovirus-based vaccines. Third, we did not evaluate neutralizing antibodies, although there is a high correlation between anti-RBD IgG levels and neutralizing antibodies[31,32], with a $r_s$ range between 0.70 and 0.78. Fourth, we could evaluate the anti-RBD antibody response against five Omicron subvariants, but not against the most recent circulating variants. Nevertheless, our findings remain relevant for understanding the impact of prime exposure on anti-RBD antibody response. Fifth, we adjusted for several confounding factors, including number of previous infections and vaccinations, time from last exposure and demographic factors, but residual confounding could be present. Sixth, T-cell response data were only available at T11, and anti-RBD antibody data for Omicron variants were limited to T10 and T11, which restricted a comprehensive longitudinal assessment and analysis of antibody kinetics. Nevertheless, we could show a consistent difference in anti-RBD IgG and IgA against Omicron variants due to first SARS-CoV-2 exposure. Finally, we analyzed a cohort of healthcare workers from the primary care sector, which provided a well-characterized cohort, but limited generalizability, including the absence of severe infections and the low proportion of males.

It is worth mentioning that we may have been underpowered for some analyses of the humoral response, particularly those examining the interaction between the first-exposure group and the number of previous exposures as well as when evaluating complete exposure history. Although we a priori set the type I error for the interaction test to 10% to increase sensitivity, simulation studies showed that this does not necessarily increase power to detect interactions[33]; thus these analyses should therefore be considered exploratory.

To conclude, we identified a higher anti-RBD antibody response against Omicron in individuals first exposed to SARS-CoV-2 antigens via vaccination than through natural infection. This difference decreased as the number of exposures increased. The antibody dynamics over time reflected in clinical protection, when first-infected had higher protection early on, while having lower protection than first-vaccinated upon Omicron arrival. In contrast to humoral response, T-cell response against Wuhan was higher in those first exposed through infection, although T-cell findings may be underpowered because of limited sample size. Our study reinforces the role of the vaccination campaigns against SARS-CoV-2, guaranteeing the first exposure to SARS-CoV-2 antigens via vaccination and resulting in an overall better immune response against Omicron.

## Methods
The study protocols were approved by the IRB Comitè Ètic d'Investigació Clínica IDIAP Jordi Gol (20/162-PCV). Written informed consent was obtained from all study participants prior to study initiation.

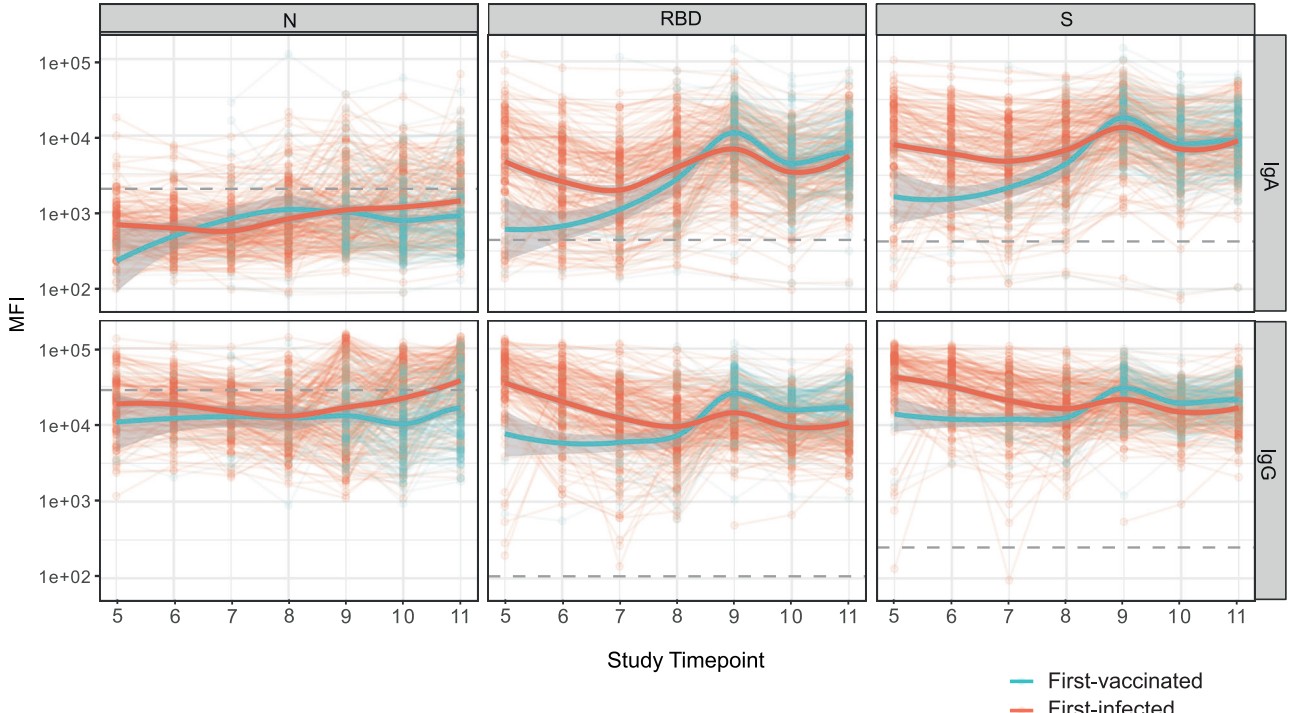

**Fig. 6 | Longitudinal Wuhan antibody levels against N, RBD, and S proteins over time.** The blue and red solid lines represent the first-vaccinated and first-infected fitted curves calculated using the LOWESS (locally estimated scatterplot smoothing, *ggplot2::geom_smooth*) method with the following parameters: degree: "2 - quadratic"; family: "gaussian"; surface: "interpolate", span 0.50. The dashed gray lines represent the mean cutoff values for each antibody and isotype to be considered positive. We used the mean cutoff across the timepoints to provide a single line in the figure. Cutoff values: For IgA, the mean cutoff values were 2018.49 (SD = 530.17) for the N, 441.30 (SD = 136.65) for RBD, and 421.53 (SD = 60.68) for S. For IgG, the mean cutoff values were 28128.12 (SD = 9968.65) for N, 103.33 (SD = 28.58) for RBD, and 246.40 (SD = 99.78) for S. Shaded areas represent 95% confidence intervals. Source data are provided in the source data. MFI: median fluorescence intensity. N: nucleocapsid. RBD: receptor binding domain. S: Spike.

## Study design, population, and setting

The CovidCatCentral cohort consists of health-care workers in primary care centers from three counties in Barcelona province, Spain, who were offered COVID-19 vaccination starting December 2020. The follow-up was conducted via eleven cross-sectional surveys, hereafter called timepoints. A first group was recruited during the first wave of the COVID-19 pandemic (March–April 2020, $n = 247$) with symptomatic SARS-CoV-2 infection confirmed by reverse transcriptase polymerase chain reaction (RT-qPCR) and/or antigen rapid diagnostic test (RDT). A second group was recruited from March–April 2021, at timepoint 5, after complete primary series vaccination ($n = 200$) and without previous SARS-CoV-2 infection, characterized by absence of a clinical episode of COVID-19 and any positive RT-qPCR, RDT, or serology tests. The second group was recruited to obtain similar characteristics (e.g., age, sex, professional category, smoking habits) to the first group. A detailed scheme for the surveys' recruitment is shown in Fig. 1.

Analyses in this study were performed with samples collected in T10 (Nov 2022) and T11 (May 2023). From 405 participants with blood samples available at T10 and/or T11, we excluded 44 (13.3%) individuals representing 51 samples for not having hybrid immunity, constituted of 29 (57%) samples with 3 previous vaccines, 10 (20%) with 2 previous infections, 5 (10%) with 4 previous vaccines, 4 (8%) with 3 previous infections, 1 sample with 4 previous infections, 1 sample with 1 previous infection, and 1 sample with 2 previous vaccines. From the remaining 361 individuals with hybrid immunity, we excluded three additional participants who had fewer than three total exposures (all with only one infection and one vaccine) and one individual with an inconsistent

exposure history between timepoints, which left 357 individuals and 646 samples (337 for T10 and 309 for T11).

Demographic and clinical data for each participant were collected at baseline and during follow-up visits through telephone interviews and electronic standardized questionnaires by study physicians and nurses as described elsewhere[19,20].

## Quantification of antibodies to SARS-CoV-2

We measured IgA and IgG plasma levels (median fluorescence intensity, MFI) to the full-length (FL) SARS-CoV-2 nucleocapsid (N) (amino acids 1-416); spike (S) antigens (amino acids 1-1213, amino acid sequence starting MFIF and ending IKWP), and the receptor binding domain (RBD) that lies within the S1 region (RBD ancestral, amino acids 319-541, amino acid sequence starting RVQP and ending CVNF; and variants Delta, BA.1, BA.2, BA.4/5, BQ.1.1, and XBB), by quantitative suspension array technology assays (xMAP, Luminex), following a previously described protocol[34–36]. A nucleotide fragment encoding the ancestral N FL, followed by a 6xHis-tag, was cloned into pET22b expression vector, transformed in *E. coli* BL21 DE3, induced with IPTG, and purified by affinity chromatography using HisTrap columns, and controlled for purity by SDS-PAGE and Coomassie staining[37]. The ancestral S FL and the RBD proteins were fused with C-terminal 6xHis and StrepTag sequences and purified from the supernatant of lentiviral-transduced CHO-S cells cultured under a fed-batch system[35].

Codon-optimized nucleotide fragments encoding the RBD from variants (Delta, BA.1, BA.2, BA.4/5, BQ.1.1, and XBB) were synthesized and cloned into a pcDNA3.1/Zeo (+) expression vector (Thermo Fisher Scientific)[38]. Recombinant proteins were produced by transient

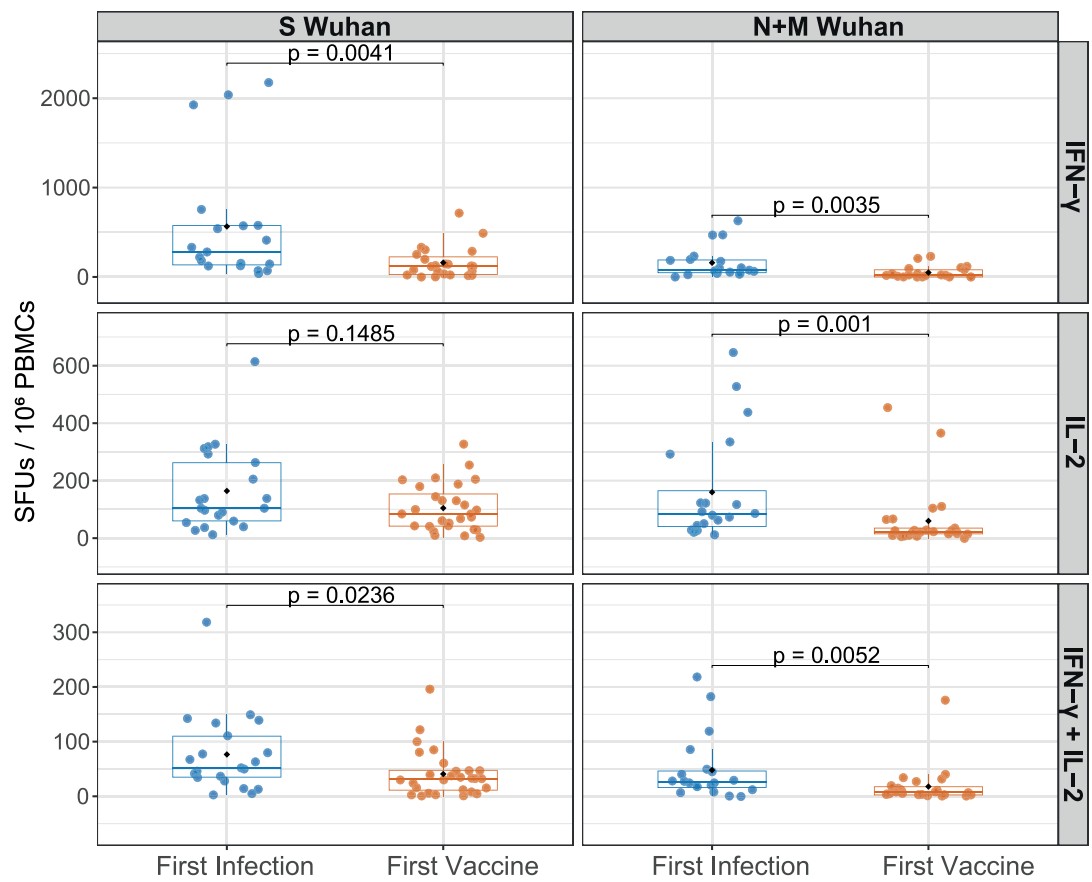

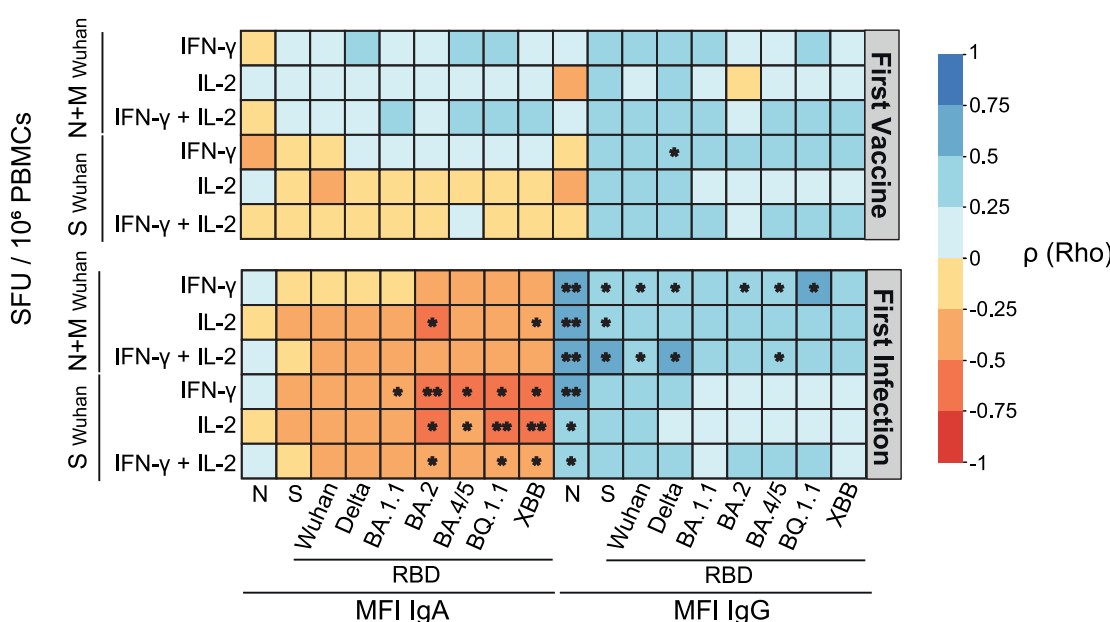

transfection of exponentially growing Freestyle TM 293-F suspension cells (Thermo Fisher Scientific) using the polyethyleneimine (PEI)-precipitation method. Proteins were purified from culture supernatants by high-performance chromatography using the Ni Sepharose® Excel Resin (GE Healthcare), according to manufacturer's instructions, dialyzed against PBS using Slide-A-Lyzer® dialysis cassettes (Thermo Fisher Scientific), quantified using NanoDrop™ One

instrument (Thermo Fisher Scientific), and controlled for purity by SDS-PAGE using NuPAGE 3-12% Bis-tris gels (Life Technologies).

Optimal testing dilutions were previously assessed to ensure samples were within the quantitative range of the assay. Plasma samples were tested at 1:500 dilution for IgA and IgG, and additionally at 1:5000 for IgG to avoid saturated anti-S IgG levels. Anti-N IgG was tested at 1:500 dilution. To quantify IgA, samples and controls were

**Fig. 7 | Magnitude of T-cell responses at T11 by first exposure groups and correlations with antibody responses. A** Boxplots representing T-cell responses as SFUs / 10⁶ peripheral blood mononuclear cells (PBMCs) secreting IFN-γ, IL-2 or IFN-γ + IL-2 (polyfunctional) to S or N + M from Wuhan by first exposure groups. Responses were compared by Wilcoxon Signed-Rank test. Boxplots represent median (bold line), the mean (black diamond), 1st and 3rd quartiles (box), and largest and smallest values within 1.5 times the interquartile range (whiskers). **B** Heatmaps illustrating the Spearman's correlation coefficient ρ (Rho) between the SFU / 10⁶

PBMCs of T-cells secreting IFN-γ, IL-2 or IFN-γ + IL-2 (polyfunctional) to S and N + M with the antibody responses (median fluorescence intensity (MFI) of IgA and IgG). *p*-values: * ≤ 0.05, ** ≤ 0.01. IFN-γ: Interferon-gamma. IL-2: Interleukin-2. M: Membrane. N: Nucleocapsid. RBD: Receptor binding domain, S: Spike. SFU: Spot-forming units. First-infected (*N* = 21), First-vaccinated (*N* = 28). P-values in both Panel **A** and **B** are from two-sided tests. Source data are provided in the source data. Note: T-cell findings may be underpowered because of limited sample size.

pretreated with anti-human IgG (Gullsorb) at 1:10 dilution, to avoid IgG interferences. Negative controls were 128 prepandemic samples from Spain and were described elsewhere[36,37].

To allow the comparison of antibody levels between T10 and T11 and correct any batch effect, we normalized the MFI values using the following formula (1):

$$normalized\ MFI = \frac{sample\ MFI}{PC\ MFI} \times mean\ MFI\ of\ T10\ and\ T11\ PCs \quad (1)$$

where PC MFI is the positive control MFI value, and Mean MFI of T10 and T11 PCs is the mean of T10 and T11 positive controls for a given isotype and antigen combination. The operators conducted the assays in a blinded manner. The results are presented as log₁₀-transformed MFI.

We evaluated the correlation between anti-RBD antibodies for Wuhan, BA.1.1, BA.4/5, and BQ1.1 with their respective neutralizing antibodies (nAb) measured in a subset of individuals at T11[39,40]. The Spearman correlation coefficient between anti-RBD antibodies and nAb ranged from 0.70 to 0.78 (Supplementary eFig. 7).

### Cellular assay
The magnitude of T-cell responses to ancestral SARS-CoV-2 FL S and N membrane (M) peptide pools [PepTivator® SARS-CoV-2 Prot_S Complete, Prot_N, Prot_M (Miltenyi)] was measured using the human IFN-γ/IL-2 FluoroSpot kit (Mabtech)[40,41] in a random subset of 49 individuals with hybrid immunity and available peripheral blood mononuclear cells (PBMCs) (21 first-infected and 28 first-vaccinated). We combined the stimuli according to vaccine or non-vaccine antigens, S and N + M, respectively.

PBMCs were isolated from venous blood samples by density-gradient centrifugation using Ficoll-Paque (Merck), cryopreserved in heat-inactivated fetal bovine serum (HI-FBS) (Thermo Fisher Scientific) with 10 % dimethyl sulfoxide (Merck), and stored in liquid nitrogen until use. After blocking the pre-coated FluoroSpot plates with culture medium-10% HI-FBS, $2.5 \times 10^5$ thawed PBMCs were added to wells containing the stimulus (1 μg/mL of peptide concentration) or the negative control (only culture medium [TexMACS Medium (Miltenyi) −1% penicillin/streptomycin (10.000 U/ml, Thermo Fisher Scientific)], and $5 \times 10^4$ PBMCs were added to the wells containing the positive control (phytohemagglutinin (Merk), 5 μg/ml). PBMCs were incubated at 37 °C and 5% CO2 for 20 h. All conditions were performed in duplicate.

Cells secreting IFN-γ and/or IL-2 were detected and counted as spot-forming units (SFU). Seven participants with ≥100 SFU in unstimulated wells for IFN-γ were excluded from the analysis. SFU counts in the unstimulated wells were subtracted from those in the stimulated wells to account for background responses, and negative values were set to zero. The results were expressed as SFUs/10⁶ PBMCs.

### Definitions
Asymptomatic infections were defined as an increase in the IgG or IgA levels between consecutive timepoints in individuals without COVID-19 symptoms and without any positive RT-qPCR or RDT or vaccination during the time interval evaluated. For those vaccinated between

timepoint intervals, an individual was considered infected if the fold change was greater than 4 for IgG or IgA against N[42]. For those not vaccinated between timepoint intervals, for the T1–T2 interval up to the T7–T8 interval, an individual was considered infected if at least two different IgG or IgA antibodies against N or S antigens exhibited a fold change greater than 4[42]. In later intervals, an individual was considered infected if at least one of the measured IgG or IgA antibodies had a fold change greater than 3. The lower fold change threshold in these later intervals was applied since individuals diagnosed with an infection during these intervals exhibited smaller fold change increases compared to earlier intervals. This reduction in fold change increase is attributed to elevated antibody levels at later timepoints. Using this approach, we were able to identify infections when compared to RT-PCR/ RDT test as follow: we detected 6/7 (85%) infections at T6-T7, 5/5 (100%) at T7-T8 39/42 (93%) at T8-T9, 68/89 (76%) at T9–T10, and 29/33 (88%) at T10–T11, supporting the validity of our antibody-based classification.

Symptomatic infections were defined by individuals with COVID-19 symptoms and a positive RT-qPCR or RDT.

Chronic comorbidities were self-reported and divided into the following categories: chronic respiratory diseases (chronic obstructive pulmonary disease, asthma), cardio-metabolic (dyslipidemia, hypertension, diabetes, other cardiovascular diseases), obesity, neurologic, gastrointestinal, chronic renal disease, immunosuppressed status (autoimmune disease, cancer, other immunosuppression), hypothyroidism, depression, pregnancy, and history of allergy. Additionally, we assessed the tobacco smoking status as active smoker, former smoker and never smoker.

The definitions of previous infections and vaccinations always followed the logic of what happened before the evaluated timepoint.

### Data analysis
We described variables as mean ± SD, median [p25-p75], and counts (%) as appropriate. We evaluated the linear correlation for the log₁₀ anti-RBD MFI values between isotypes, the seven RBD from variants, and T10 and T11 with the Pearson correlation coefficient, after evaluating their normal distribution graphically.

We compared the anti-RBD antibody levels between the first-vaccinated individuals with the first-infected ones (reference group) at T10 and T11 using linear mixed models. The models accounted for the repeated sampling with random intercepts per individual. We defined a priori a set of potential confounding variables and performed a sequential adjustment as follows: M0, without adjustment; M1, M0 plus age (restricted cubic spline with 3 degrees of freedom (df)) and sex; M2, M1 plus number of chronic comorbidities and tobacco smoking status; M3, M2 plus number of non-Omicron and Omicron symptomatic infections (as factors), and number of non-Omicron and Omicron asymptomatic infections (as factors); M4, M3 plus number of non-bivalent and bivalent vaccines (as factors); and finally, M5 (main model), M4 plus days from last infection (restricted cubic spline with 3 df) and days since last vaccine (restricted cubic spline with 3 df).

We ran four sensitivity analyses: MS1, using M5 (main model) but expanding the number of chronic comorbidities as three binary factors (cardio-metabolic, immunosuppressed and previous allergy); MS2, running M5 (main model) in those individuals without any history of

asymptomatic infection; MS3, using M5 (main model) but instead of non-bivalent/bivalent, using number of previous mRNA and adenovirus based vaccines; and MS4, running MS3 but instead of number of non-omicron/omicron, using number of previous infections by each variant. We further evaluated the same models using the anti-RBD/anti-S antibody ratio as the outcome, as a potential better surrogate for nAb. A summary of the model's adjustment is in Supplementary eTable 4. In the previous models, we did not consider any interaction term. Stratification by sex was considered during statistical planning but was not performed due to the large predominance of females, leading to unbalanced group sizes.

Based on the biological rationale and literature, we fitted the main model (M5) with an interaction term between the exposure (first-vaccinated x first-infected) and the number of previous exposures (as factor) to evaluate whether the difference between first-infected and first-vaccinated depends on the total number of previous exposures. We evaluated a potential statistical interaction with a likelihood ratio test comparing the model with and without the interaction term. This analysis was conducted among those with a minimum of 3 and maximum of 6 previous exposures, because only those in the first-infected group had more than 6 previous exposures ($n = 3$ individuals).

To evaluate whether the difference between first-infected and first-vaccinated occurs mainly due to specific previous vaccination and infection histories, i.e., the exact sequence of exposures, we fitted a model adjusted as M5 but using the exact sequence of exposures instead of number of previous infections and vaccinations. To create this sequence, we considered asymptomatic and symptomatic infections as equal previous infections and non-bivalent and bivalent vaccines as equal previous vaccines, to deal with sparse data and improve power. We fitted the model with a categorical variable containing the 9 most common history combinations (i.e., combinations with at least 20 samples, $n = 212$ individuals), while collapsing those below 20 samples as "others" ($n = 145$ individuals). Among the 9 most common history combinations, we chose those with a meaningful contrast to understand the first exposure effect (i.e., those with the same number of exposures, but differing on the prime-exposure), resulting in eight sequences and four comparisons. The confidence intervals for these contrasts accounted for Bonferroni corrections. In a sensitivity analysis, we evaluate these comparisons without multiplicity correction and using false-discovery rate (FDR) method.

The results from the linear mixed models are presented as a transformed beta value with the formula: $([10^{\wedge}\beta]-1)*100$, giving the difference (in percentage) in antibody levels when comparing first-vaccinated to the first-infected (reference group). We estimated the 95% confidence intervals (CI) for the estimates from the model. All linear mixed models were evaluated regarding residual diagnostics and multicollinearity with variance inflation factors (VIF). All models were within the expected assumptions and there was no evidence for multicollinearity (i.e., VIF < 5). The VIF values for the main model (M5) for all analytes and isotypes are shown in Supplementary eFig. 8.

To explore the observed difference between first-vaccinated and first-infected, we evaluated antibody kinetics for IgA and IgG against Wuhan with anti-S, anti-N and anti-RBD antibodies using the data available from T5 to T11. We plotted the MFI levels against each timepoint and fitted a locally weighted scatterplot smoothing (LOWESS) curve. The LOWESS was fit with the ggplot2::geom_smooth function with the following parameters: degree: "2 - quadratic"; family: "gaussian"; surface: "interpolate", span 0.50. To evaluate another method to account for the nonlinearity, we modeled it using a generalized additive model (GAM), with the mgcv package, and let its robust algorithm[43] automatically set the non-linear thin-plate spline. We also performed different parametrizations, as sensitivity analyses, to analyze whether the smoothing itself was driving any of the observed differences (Supplementary eTable 5).

To investigate the association between the first exposure group and the risk of SARS-CoV-2 breakthrough infections, we conducted a time-to-event analysis using Cox proportional hazards regression modeling. We used three sub-cohorts starting at timepoints T8, T9 and T10, with individuals being followed up to the starting point of the next timepoint (i.e., T9, T10 and T11). Individuals were followed up until the first episode of SARS-CoV-2 breakthrough infection (outcome of interest), receipt of an additional vaccine dose, or the last day of study follow-up (end of T9, T10, or T11, respectively). The main model included both asymptomatic and symptomatic infections as outcome of interest. The date of asymptomatic infections was inferred by taking the midpoint between the timepoints defining the period. To avoid misclassification, participants who had both an asymptomatic infection and a vaccination within the same timepoint interval were excluded from the analysis. As a sensitivity analysis, we ran the same Cox model considering only symptomatic infections. Models were adjusted by age, sex, number of comorbidities, smoking status, number of previous exposures, and time since last exposure. The Cox proportional hazards assumption was evaluated by examining Schoenfeld residuals.

Nonparametric tests were used to analyze T-cell data. We compared responses to S and N + M by paired Wilcoxon signed-rank tests. The proportions (%) of secreting T cells induced by S vs. N + M antigens were compared using the chi-square test. Comparisons of the magnitude of T-cell responses between first-infected/first-vaccinated groups were performed by the Wilcoxon rank-sum test.

A P-value of ≤0.05 was considered statistically significant for all estimates, except for the interaction between the first-group exposure and number of previous events when we considered a $P \le 0.10$. We decided a priori to increase the type 1 error rate to 10% because of the expected low power to test for interaction in this scenario[44,45], particularly considering a $2 \times 4$ categorical interaction terms in the model. We performed the statistical analysis in the software R version 4.2.1.

## Reporting summary

Further information on research design is available in the Nature Portfolio Reporting Summary linked to this article.

## Data availability

In accordance with the current European and national law, the data used in this study are only available for the researchers participating in this study. Thus, we are not allowed to distribute or make publicly available the data to other parties. Researchers wishing to obtain the data must submit a request to the corresponding data custodian (Prof. Carlota Dobaño; carlota.dobano@isglobal.org). Requests will be evaluated within one month. Access will be granted only for research purposes that are compatible with the original ethics approval and will require (i) evidence of institutional affiliation, (ii) a data-use agreement, and (iii) approval from a relevant ethics committee if newly mandated by the proposed use. All shared data will be provided in de-identified form, and use will be restricted to the purposes agreed upon in the data-use agreement. Source data are provided in the source data file. Source data are provided with this paper.

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

## Acknowledgements

The authors would like to express their gratitude to the participants of this study for their time and involvement. We also thank Eduard Velasco at IDIAP JG for the administrative support and the nursing team for their assistance with sample collection. We thank Selena Alonso, Natalia Diaz, Robert Mitchell, Chenjerai Jairoce and Laura Puyol at ISGlobal for laboratory support; Hugo Rozas for data management; Miriam Ramírez

(ISGlobal), Daniel Parras (IDIBAPS), Carlo Carolis and Natalia Rodrigo Melero (CRG), for antigen procurement; Omicron RBD-encoding plasmids were donated by Hugo Mouquet (Institut Pasteur). We thank Edwards Pradenas, Benjamin Trinité and Julià Blanco (IRSICaixa) for nAbs data. This work was supported by the European Union under grant agreement no. 101046314 (END-VOC) and by the Fundació Privada Daniel Bravo Andreu. CM was supported by the 100046TC21_ 2022 INV-1 00046 grant funded 535 by AGAUR/European Union NextGenerationEU/PRTR. OR is funded by the Ramón y Cajal program (RYC2023-002923-C) awarded by the Spanish Ministry of Science, Innovation and Universities (MICIU/AEI/10.13039/501100011033) and by the European Social Fund Plus (ESF +). RR had the support of the Health Department, Catalan Government (PERIS SLT017/20/000224). GM was supported by RYC 2020-029886-I/AEI/10.13039/501100011033, co-funded by European Social Fund (ESF). PS was supported by PID2021-125493OB-I00 from MICIN. We acknowledge support from the grant CEX2023-0001290-S funded by MCIN/AEI/ 10.13039/501100011033, and support from the Generalitat de Catalunya through the CERCA Program.

## Author contributions

G.M., A.R.-C., A.R.-M., J.V.-A., and C.D. designed the cohort study. A.R.-M., J.V.-A. and A.R.-C. recruited participants, collected data and obtained samples. A.J., M.V., D.B., M.C., R.R., and I.C. processed the samples, developed and/or performed the antibody binding assays and data preprocessing. R.R., M.C. and C.T. performed cellular immunology analyses. L.I., L.M.-A., R.R. and P. Santamaria and P. Serra contributed with key reagents/antigens and expertise. O.R. and C.M.P. analyzed and/or managed the data. G.M., R.A. and C.D. supervised the antibody assays and data analyses. O.R. wrote the first draft of the paper with contributions from R.R., C.M.P., G.M. and C.D. All authors reviewed and approved the final version as submitted to the journal.

## Competing interests

P. Santamaria is founder, scientific officer and stockholder of Parvus Therapeutics and receives funding from the company. He also has a consulting agreement with Sanofi. The other authors declare no competing interests.

## Additional information

[1]ISGlobal, Barcelona, Catalonia, Spain. [2]DataHealth Lab, Institut de Recerca Sant Pau (IR SANT PAU), Barcelona, Spain. [3]Facultat de Medicina i Ciències de la Salut, Universitat de Barcelona (UB), Barcelona, Spain. [4]Unitat de Suport a la Recerca de la Catalunya Central, Fundació Institut Universitari per a la recerca a l'Atenció Primària de Salut Jordi Gol i Gurina (IDIAP JG), Manresa, Spain. [5]CIBER Epidemiologia y Salud Pública (CIBERESP), Barcelona, Spain. [6]Centro de Investigação em Saúde de Manhiça (CISM), Maputo, Mozambique. [7]Institut d'Investigacions Biomèdiques August Pi Sunyer, Barcelona, Spain. [8]Department of Microbiology, Immunology and Infectious Diseases, Snyder Institute for Chronic Diseases, Cumming School of Medicine, University of Calgary, Calgary, AB, Canada. [9]CIBER de Enfermedades Infecciosas (CIBERINFEC), Instituto de Salud Carlos III, Barcelona, Spain. [10]Health Promotion in Rural Areas Research Group (PROSAARU), Gerència d'Atenció Primària i a la Comunitat Catalunya Central, Institut Català de la Salut, Manresa, Spain. [11]Faculty of Medicine, Universitat de Vic-Universitat Central de Catalunya (UVIC-UCC), Barcelona, Spain. [12]Centre d'Atenció Primària (CAP) Sant Joan de Vilatorrada, Institut Català de la Salut, Sant Joan de Vilatorrada, Spain. [13]These authors contributed equally: Otavio Ranzani, Carla Martín Pérez. [14]These authors jointly supervised this work: Gemma Moncunill, Carlota Dobaño. ✉e-mail: oranzani@santpau.cat; carlota.dobano@isglobal.org

