## [Peer Review File · Nature Communications]

Primary SARS-CoV-2 Exposure by Vaccination or Infection Shapes Immune Responses to Omicron Variants among a Spanish Cohort

Corresponding Author: Dr Otavio Ranzani

Version 0:

Reviewer comments:

Reviewer #1

(Remarks to the Author)

General Comments

In this study by Ranzani et al. address whether the initial priming acquired through either infection or vaccination affects the resulting immune response later on against SARS-CoV-2 Omicron variants in the setting of hybrid immunity. They analyzed the antibody binding and T-cell responses at T10 and T11 time points (i.e. Omicron BQ.1.1 and XBB variants and recent ones) in a longitudinal cohort (CoviCatCentral) of health-care workers (n=350+) with blood sampling spanning 2020-2023.

They found a higher anti-RBD antibody levels against Omicron in individuals initially exposed to SARS-CoV-2 antigens through vaccination compared to those first exposed through natural infection. This difference wanes with an increasing number of exposures. Further, the antibody levels over time correlated with clinical protection: those first-infected had higher protection early on, whereas those first-vaccinated showed greater protection later. However, the T-cell response was higher in individuals first exposed through infection.

It is understood that studying antibody responses is critical in infection vs vaccination in hybrid immune cohorts for future vaccination strategies and supporting the ongoing research. The authors have access to an excellent longitudinal cohort. However, there are few critical details in the current manuscript that are unclear, with significant concerns that needs to be addressed. Overall, there is a lack of presentation of the actual data, with only statistical analyses presented. Presentation of data is relatively poor, it is very confusing and lacking in detail. The key message that hybrid immune folks with an initial vaccine priming shows higher titers at late stages is significantly muddled by mismatched demographics and underlying co-factors, as well as the fact that the vaccine primed group had more breakthrough infections at TP 8 and 9, making the main finding fully expected. Finally, without information on neutralizing titers the dataset is very difficult to fully interpret. The key message that can be conveyed from this well-characterized cohort is poorly conveyed.

Specific Comments

1. Unfortunately, there is no clarity about the source of mRNA vaccine received by the study participants. Was it Pfizer or Moderna or a mix of both? If they got mRNA vaccine from multiple sources, then it would be better to characterize the immune response comparing both vaccines.
2. The details of SARS-CoV-2 antigenic exposures in terms of breakthrough infections caused by specific distinct variants is missing and data is not analyzed keeping in mind the impact of different variants experienced through breakthrough infections. So, it will be critical to describe the impact of breakthrough infection early during the Omicron wave on B and T cell responses at T10 and T11 time points.
3. Many donors were removed from consideration based on no clearly defined criteria.
4. The actual longitudinal titers are poorly presented. In figure 5, longitudinal data are shown, but it is not even made clear in the legend what variant was used. Why are the N specific titers so high for all donors? What negative controls were used for these analyses?
5. BQ.1.1 and XBB are currently outdated and no data is shown against the recently or currently circulating variants in this study.

6. The absence of neutralization data makes this study weaker. There are no experiments done that supports the statements about protective antibodies. Just RBD binding antibodies can't be called as protective as all RBD antibodies are not always able to neutralize the virus.
7. Correlation analysis in Supp Fig 2. The Pearson's correlation test was applied. However, there is no information whether the data is normally distributed or parametric. If this is not the case, then Spearman's correlation should be applied. Please provide the details.
8. Figure 1 and Supp Fig. 1 should be combined for better understanding.
9. Lines 306-307: It is unclear about the sampling time point for the following, 'and it was shorter for the first- vaccinated group compared to the first-infected group for T10 and similar for T11'.
10. Figure legends are missing key information to be interpretable.
11. Figure 2 states it is a comparison of differences of antibody responses in the donors primed by infection and vaccination. It is unclear how this is represented in this data.
12. How was the assay they use validated? Especially the N specific assay which shows high titers for all donors at TP 5.

Minor

1. Please provide sequence details of AA/number positions for S, RBD and N antigens used in this study. Further, it would be great to include their QC data e.g. SDS-PAGE and Native-PAGE gel images.
2. Supp Table 1: Table is incomplete on the right side for IgA response.

Reviewer #2

(Remarks to the Author)

Reviewer #3

(Remarks to the Author)

In their manuscript, Ranzani and colleagues analyze antibody responses to different parts of SARS-CoV-2 and its variant over time and multiple exposures. The work is well done and the manuscript is interesting. But there are some points that need the authors' attention.

Major points

- 1) Parts of the manuscript are a little hard to follow. The authors should make an effort to make it easier accessible for readers.
- 2) While the work adds to our knowledge of the evolution of immune responses to SARS-CoV-2 over time, it is descriptive in nature and does not provide mechanistic insights.
- 3) It has been reported that ratios between RBD and spike antibodies change over time. The authors should analyze this (since the titers have already been measured anyways).

Minor points

- 1) Many abbreviations, including in the abstract are not defined. E.g. COVID-19, RBD, SARS-CoV-2 in the abstract. Main body: COVID-19, IRB, IPTG, SDS-PAGE, CHO, SD,
- 2) Line 75-76: Is this the fact? There are several papers showing much higher peak antibody responses after initial vaccination than after initial infection. Please check the literature and add references to support that statement.
- 3) Line 80, line 249: It should be 'prime-exposure'.
- 4) Line 121: 'which left', instead of 'remaining'.
- 5) Line 131, line 260, line 275, line 347: Please do not use capitalized letters to start words mid-sentence. It should be 'spike', receptor binding domain' etc.
- 6) Line 141: 'into a'
- 7) Line 157: 'batch'
- 8) Line 179: Please specify the actual penicillin and streptomycin concentrations.

9) Line 248: 'effecy'?

10) Line 288: 'vaccinated'

11) Line 354: 'infections'

12) 'et al.' should always be in italics.

Reviewer #4

(Remarks to the Author)

This manuscript tried to addresses a critical public health topic of the antibody and cellular responses by investigating the effects of different initial exposure routes on immune responses. The manuscript presents interesting findings on the immune response dynamics based on the first exposure route to SARS-CoV-2 antigens. However, the significant methodological and interpretative issues undermine the reliability of the conclusions. The authors are encouraged to address the following issues comprehensively to improve the rigor and reliability of the manuscript.

Study Design and Sample Limitations

1. Sample Size and Distribution Imbalance

- 1) 、 The sample sizes of the first-infected group (n=197) and the first-vaccinated group (n=160) are unbalanced, potentially affecting statistical power.
- 2) 、 The cellular assay is based on only 49 individuals with hybrid immunity, which is insufficient for robust subgroup comparisons (first-infected vs. first-vaccinated).
- 3) 、 Please provide a justification for the current sample size and its adequacy, or expand the sample size, particularly for highly exposed groups, to ensure statistical reliability.

2. Timeframe and Virus Strain Variability

- 1) 、 The two groups had their first exposure in 2020 and 2021, involving different viral strains (Wuhan vs. Alpha). This undermines the comparability of the groups, as different strains may trigger distinct immune responses. The differences between the Wuhan and Alpha strains likely contributed to variations in antibody responses, further complicating the comparison.
- 2) 、 The intervals between the last exposure and sampling (T10 vs. T11) differ between the groups, potentially affecting the observed antibody decay rates.
- 3) 、 Please verify the data in Appendix Figure 1 against the main text, with particular attention to the timeframe and sample size information.

3. Quantification of Exposure History

- 1) 、 The methods do not clearly explain how the sequence of exposures (e.g., infection followed by vaccination) was quantified or incorporated into the analysis.
- 2) 、 Please provide a detailed explanation of how exposure sequences were quantified and their influence on immune responses.

Statistical Analysis Issues

1. Complexity and Potential Bias in Statistical Models:

- 1) The main model (M5) incorporates a large number of confounding variables, raising concerns about overfitting and multicollinearity (e.g., between vaccine doses and time intervals).
- 2) High exposure counts are only observed in the first-infected group, creating an imbalance for interaction effect analysis.
- 3) Simplify the statistical model by reducing unnecessary variables. Please provide diagnostic results (e.g., AIC/BIC, residual analysis) to validate model quality.

2. Conservative Correction for Multiple Comparisons

- 1) Bonferroni correction, while controlling type I error, is overly conservative and risks type II errors (false negatives).
- 2) Present results using both Bonferroni and False Discovery Rate (FDR) corrections, and compare unadjusted and adjusted p-values to evaluate their impact.

3. Fitting of Antibody Dynamics

- 1) The LOWESS curve used for antibody dynamics is prone to edge effects, potentially leading to inaccuracies at the endpoints.
- 2) Provide details on LOWESS parameters and perform sensitivity analyses to verify the robustness of the fits. Consider alternative models, such as generalized additive models (GAM), for better reliability.

4. p-value

- 1) The manuscript applies different p-value thresholds for main effects (≤ 0.05) and interaction terms (≤ 0.10). Could the authors clarify the rationale behind using a less stringent threshold for the interaction term and discuss the potential implications of this choice on the interpretation of the results?

Results and Interpretation

1.Lack of Dynamic Analysis:

- 1) The study only compares T10 and T11 timepoints, missing insights into the dynamic trends of immune responses over time.
- 2) Incorporate longitudinal analyses to better characterize antibody decay rates and T-cell dynamics.

2.Inadequate Adjustment for Confounding Factors:

- 1) Significant group differences exist in age ($p = 0.006$) and comorbidities (e.g., cardio-metabolic conditions, $p = 0.016$, and allergy history, $p = 0.024$), which may confound immune response comparisons.
- 2) Either adjust for these confounders in statistical models or use propensity score matching to balance the groups.

3.Unclear Definitions of Asymptomatic Infections:

- 1) Asymptomatic infections are inferred solely based on antibody level changes, which could be influenced by other factors (e.g., vaccination, individual immune traits), leading to potential misclassification.
- 2) Use additional evidence (e.g., PCR results) to corroborate the classification of asymptomatic infections.

Minor points

- 1.In the formula " $[10^{\beta} - 1] * 100$ ", " β " should use the correct Greek symbol with proper spacing.
- 2."effecy" is a spelling error and should be corrected to "effect."
- 3.The term "non bivalent" should be written as "non-bivalent" consistently throughout the manuscript.
- 4.When LOWESS is introduced, its full name ("Locally Weighted Scatterplot Smoothing") should be provided.
- 5.Inconsistent formatting of terms like "anti-RBDWuhan" should be corrected to "anti-RBD Wuhan."
- 6."...with individuals being followed-up up to the starting point...", the repeated "up" should be corrected by removing one instance.

Version 1:

Reviewer comments:

Reviewer #1

(Remarks to the Author)

The current version of the manuscript is substantially improved. I am content with the changes made and have no further concerns.

Reviewer #4

(Remarks to the Author)

I appreciate the authors' thorough response to the first-round review comments and their revisions, and I commend the improvements made in the clarity and structure of the manuscript. This study addresses a timely and important question regarding how the sequence of SARS-CoV-2 exposures influences the evolution of immune responses. However, there remain some critical issues in the interpretation of results and statistical methods that require further clarification and strengthening. I encourage the authors to carefully consider the following suggestions to enhance the scientific rigor and persuasiveness of the manuscript. The detailed review comments are as follows.

1.The subgroup of 49 individuals used for T-cell analysis appears underpowered to support robust comparisons between primary infection and primary vaccination. Although this limitation is acknowledged in the discussion, it should also be clearly stated in the abstract and conclusion. Descriptions such as "consistent patterns" may overstate the certainty of the findings and should be avoided or qualified with appropriate language. Please also add relevant cautionary notes in figure legends and the methods section.

2.If feasible, report statistical power or confidence intervals for subgroup analyses to help readers assess the reliability of the findings. If not possible, a brief note discussing the limitations of statistical power in the methods or discussion would be helpful.

3.The modeling framework remains insufficiently detailed. Although the authors provided a table summarizing covariate adjustments across different models, it is unclear whether interaction terms were included or evaluated. Please clarify if any interaction analyses were performed. If not, the authors should consider conducting interaction analyses where appropriate, such as when there is biological rationale or prior evidence suggesting potential interactions.

4.To address concerns about multicollinearity, consider including a summary of variance inflation factor (VIF) values, either in the main text or supplementary materials. It would also be helpful to share full regression outputs—including coefficients and p-values—so that readers can directly assess the results.

5.The manuscript mentions FDR correction, but it is unclear whether the primary findings remain statistically significant after adjustment. Please state this explicitly in the main text, and consider using visual indicators (e.g., color or symbols) in figures to highlight adjusted significance levels.

6.The observed antibody trends are derived using LOWESS smoothing, but no quantitative or visual evidence is provided to rule out artifacts from boundary effects. A more systematic evaluation is needed to demonstrate that the observed trajectories

are robust and not driven by the smoothing procedure itself.

7. A relaxed significance threshold ($p \leq 0.10$) is used for interaction terms without prior specification or justification. While reduced power for interactions is a known issue, applying a looser threshold post hoc without a pre-defined rationale may introduce bias. Consider using a consistent threshold (e.g., $p \leq 0.05$) or providing a formal justification (e.g., through power analysis or pre-specified plans) for differential treatment.

Version 2:

Reviewer comments:

Reviewer #4

(Remarks to the Author)

I appreciate the authors' thoughtful revisions. I have no additional concerns.

REVIEWER COMMENTS

Reviewer #1 (Remarks to the Author):

General Comments

In this study by Ranzani et al. address whether the initial priming acquired through either infection or vaccination affects the resulting immune response later on against SARS-CoV-2 Omicron variants in the setting of hybrid immunity. They analyzed the antibody binding and T-cell responses at T10 and T11 time points (i.e. Omicron BQ.1.1 and XBB variants and recent ones) in a longitudinal cohort (CoviCatCentral) of health-care workers (n=350+) with blood sampling spanning 2020-2023.

They found a higher anti-RBD antibody levels against Omicron in individuals initially exposed to SARS-CoV-2 antigens through vaccination compared to those first exposed through natural infection. This difference wanes with an increasing number of exposures. Further, the antibody levels over time correlated with clinical protection: those first-infected had higher protection early on, whereas those first-vaccinated showed greater protection later. However, the T-cell response was higher in individuals first exposed through infection.

It is understood that studying antibody responses is critical in infection vs vaccination in hybrid immune cohorts for future vaccination strategies and supporting the ongoing research. The authors have access to an excellent longitudinal cohort. However, there are few critical details in the current manuscript that are unclear, with significant concerns that needs to be addressed. Overall, there is a lack of presentation of the actual data, with only statistical analyses presented. Presentation of data is relatively poor, it is very confusing and lacking in detail. The key message that hybrid immune folks with an initial vaccine priming shows higher titers at late stages is significantly muddled by mismatched demographics and underlying co-factors, as well as the fact that the vaccine primed group had more breakthrough infections at TP 8 and 9, making the main finding fully expected. Finally, without information on neutralizing titers the dataset is very difficult to fully interpret. The key message that can be conveyed from this well-characterized cohort is poorly conveyed.

ANSWER: Thank you for your comments.

Specific Comments

1. Unfortunately, there is no clarity about the source of mRNA vaccine received by the study participants. Was it Pfizer or Moderna or a mix of both? If they got mRNA vaccine from multiple sources, then it would be better to characterize the immune response comparing both vaccines.

ANSWER: We have now retrieved the type of vaccines from the cohort. Overall, the main vaccine used was Pfizer (83% of the entire number of vaccines). For the first vaccine dose, it was 100% Pfizer for the first-vaccinated group and 94% for the first-infected group.

Figure 2. The entire history of previous infections and previous vaccines stratified by first-infected and first-vaccinated groups

* There was only 1 Janssen that we grouped with Astrazeneca as Adeno-virus based vaccine.

Since the vast majority of vaccinations were with Pfizer, we could not i) evaluate the influence of the first vaccine-type on our main research question (whether first-vaccinated x first-infected); ii) the impact of subsequent vaccine types, as they were too few cases to analyze r Pfizer x Moderna x Adeno-virus vaccines in the model. In one of the new sensitivity analyses, we adjusted by “number of previous mRNA vaccines” and “number of previous adenovirus-based vaccines”. We have added the above figure to the main text (new **Figure 2**).

The most administered vaccine type was Pfizer (n=840, 83%), followed by Moderna (n=162, 16%), Astrazeneca (n=10, 1%) and Janseen (n=1, 0.1%). Regarding the first vaccine exposure, Pfizer was 100% for those first-vaccinated and 94% for those first-infected. The number of

bivalent vaccines was similar between groups. The distribution of the history of previous infections and vaccines types are shown in Figure 2.

2. The details of SARS-CoV-2 antigenic exposures in terms of breakthrough infections caused by specific distinct variants is missing and data is not analyzed keeping in mind the impact of different variants experienced through breakthrough infections. So, it will be critical to describe the impact of breakthrough infection early during the Omicron wave on B and T cell responses at T10 and T11 time points.

ANSWER: We have now derived for each infection the likely VoC based on the period dominance of VoC in Catalonia, using data from <https://sivic.salut.gencat.cat> and Covidtag. We described the VoCs and accounted for them in the new sensitivity analyses, models MS3 and MS4. Originally, we assumed that the major shift in immune escape and humoral response was driven by Omicron. Thus, we considered only Omicron vs non-Omicron infections in the first submission in the model adjustment.

These are our new first-vaccinated minus first-infected estimates for anti-RBDs antibodies. The main model is Model 5 (M5 in black). The two new sensitivity analyses are MS3 (as Model 5, but replacing the number of non-Omicron/Omicron previous infections by number of previous infections by each one of VoC and its subvariants), and MS4 (as MS3, but replacing the number of non-bivalent/bivalent vaccines by number of previous mRNA and previous adenovirus based vaccines). Models results are shown below (new **Figure 3**). The description of each model adjustment is shown in the methods and eTable 1.

Figure 3. Difference on IgA and IgG antibody responses to RBD at T10 and T11 (first-vaccinated group minus first-infected group)*

* Estimates from linear mixed models estimating the %MFI increase in those first-vaccinated compared to the first-infected group (reference). The models accounted for repeated measurements in the same individual with a random intercept per individual. The models were adjusted as follows: M0, crude; M1, M0 + age (restricted cubic spline with 3df) and sex; M2, M1 + number of chronic comorbidities and tobacco smoking status; M3, M2 + number of non-Omicron and Omicron symptomatic infections (as factors), and number of non-Omicron and Omicron asymptomatic infections (as factors); M4, M3 + number of non-bivalent and bivalent vaccines (as factors); and finally, M5 (main model), M4 + days from last infection (restricted cubic spline with 3df) and days from last vaccine (restricted cubic spline with 3df). MS1, M5 but expanding the number of chronic comorbidities as three binary factors (cardio-metabolic, immunosuppressed and previous allergy); MS2, running M5 in those individuals without any history of asymptomatic infections; MS3, M5 but instead of non-bivalent/valent, using number of mRNA and adeno-virus based vaccines; MS4, MS3 but instead of number of non-omicron/omicron, using number of previous infections by each VoC.

We do not have T-cell response data for timepoint 10, and for T11, we observed that participants infected with Wuhan/Delta or Wuhan/Delta and Omicron exhibited higher T-cell magnitude compared to those infected only with Omicron. Nevertheless, when stratifying by specific Omicron subvariants, the sample size was too small to draw definitive conclusions.

3. Many donors were removed from consideration based on no clearly defined criteria.

ANSWER: Thank you for the opportunity to clarify the inclusion/exclusion criteria, which were not sufficiently detailed in the original manuscript.

Our study specifically focused on individuals with hybrid immunity (at least one documented infection and at least one vaccination) and a minimum of three exposures in total. This selection strategy was implemented for several critical reasons. First, by requiring both infection- and vaccine-induced immunity in all participants, we could more accurately compare how the type of initial exposure influenced subsequent anti-RBD responses to Omicron, which was the primary question. Second, this approach helped control for potential unmeasured confounding factors like differential exposure risks or health behaviors that might differ between those with purely natural versus purely vaccine-induced immunity. Third, current evidence suggests antibody responses reach maximal breadth after ~3 exposures; thus, selecting those with at least 3 exposures helps minimize variability

from other sources than the exposure of interest and the confounders we have available to adjust for.

From 405 participants with blood samples available at T10 and/or T11, we excluded 44 (13.3%) individuals representing 51 samples for not having hybrid immunity, constituted of 29 (57%) samples with 3 previous vaccines, 10 (20%) with 2 previous infections, 5 (10%) with 4 previous vaccines, 4 (8%) with 3 previous infections, one sample with 4 previous infections, one sample with 1 previous infection, and one sample with 2 previous vaccines. From the remaining 361 individuals with hybrid immunity, we excluded three additional participants who had fewer than three total exposures (all with just one infection and one vaccine) and one individual with an inconsistent exposure history between timepoints. We clarified it in the manuscript (Study design, population and setting Section).

Analyses in this study were performed with samples collected in T10 (Nov 2022) and T11

(May 2023). From 405 participants with blood samples available at T10 and/or T11, we excluded 44 (13.3%) individuals representing 51 samples for not having hybrid immunity, constituted of 29 (57%) samples with 3 previous vaccines, 10 (20%) with 2 previous infections, 5 (10%) with 4 previous vaccines, 4 (8%) with 3 previous infections, 1 sample with 4 previous infections, 1 sample with 1 previous infection, and 1 sample with 2 previous vaccines. From the remaining 361 individuals with hybrid immunity, we excluded three additional participants who had fewer than three total exposures (all with only one infection and one vaccine) and one individual with an inconsistent exposure history between timepoints which left 357 individuals and 646 samples (337 for T10 and 309 for T11).-

4. The actual longitudinal titers are poorly presented. In figure 5, longitudinal data are shown, but it is not even made clear in the legend what variant was used. Why are the N specific titers so high for all donors? What negative controls were used for these analyses?

ANSWER: We agree with the reviewer and we have improved the presentation of our previous Figure 5 (current **Figure 6**). The longitudinal data presented referred to antibodies against Wuhan. We would like to remember that the main analysis of our manuscript is based on the T10 and T11 evaluations of anti-RBDs antibodies against Omicron.

Please, see below some clarifications for the questions:

- The data shown in (current) Figure 6 refers to antibodies against Wuhan only, which we clarified in the new revised version;
- The dilution of anti-N antibodies was 1/500, and not 1/5000 as for Spike antigens (IgG), which we clarified in the new revised version;
- Negative controls were 128 prepandemic samples from Spain;
- The anti-N assay was developed at the beginning of the pandemic in our Lab. Performance of the assay can be found in these references.^{1,2}

Figure 6. Longitudinal Wuhan antibody levels against nucleocapsid (N), receptor-binding domain (RBD), and spike (S) proteins over time

The blue and red solid lines represent the first-vaccinated and first-infected fitted curves calculated using the LOWESS (locally estimated scatterplot smoothing, *ggplot2::geom_smooth*) method with the following parameters: degree: “2 - quadratic”; family: “gaussian”; surface: “interpolate”, span 0.50. The dashed gray lines represent the mean cutoff values for each antibody and isotype to be considered positive. We used the mean cutoff across the timepoints to provide a single line in the figure. Cutoff values: For IgA, the mean cutoff values were 2018.49 (SD = 530.17) for N, 441.30 (SD = 136.65) for RBD, and 421.53 (SD = 60.68) for S. For IgG, the mean cutoff values were 28128.12 (SD = 9968.65) for N, 103.33 (SD = 28.58) for RBD, and 246.40 (SD = 99.78) for S. Shaded areas represent 95% confidence intervals.

IgG, and additionally at 1:5000 for IgG to avoid saturated anti-S IgG levels. Anti-N IgG was tested at 1:500 dilution. To quantify IgA, samples and controls were pretreated with anti-human IgG (Gullisorb) at 1:10 dilution, to avoid IgG interferences.

Negative controls were 128 prepandemic samples from Spain and were described elsewhere.^{23,24}

5. BQ.1.1 and XBB are currently outdated and no data is shown against the recently or currently circulating variants in this study.

ANSWER: Thank you for highlighting this point. Unfortunately, we did not have data for the most recent SARS-CoV-2 variants. Nevertheless, our study is one of the first comprehensive investigations into how initial SARS-CoV-2 exposure shapes subsequent immune responses to variants, including five Omicron lineages. We have now acknowledged the absence of data on newer variants as a limitation. Despite this, our findings remain relevant for understanding the impact of prime exposure on COVID-19 immunity.

Third, we did not evaluate neutralizing antibodies, although there is a high correlation between anti-RBD IgG levels and neutralizing antibodies^{42,43} with a r_s range between 0.70 and 0.78. -Fourth, we could evaluate the anti-RBD antibody response against five Omicron subvariants, but not against the most recent circulating variants. Nevertheless, our findings remain relevant for understanding the impact of prime exposure on anti-RBD antibody response.

6. The absence of neutralization data makes this study weaker. There are no experiments done that supports the statements about protective antibodies. Just RBD binding antibodies can't be called as protective as all RBD antibodies are not always able to neutralize the virus.

ANSWER: We agree this is a limitation, nevertheless several studies have shown a strong correlation between RBD antibodies and NAb levels for COVID-19,³⁻⁶ including using our assays in our cohort and another conducted in a German cohort.⁷ Our group, as others, also showed the level of these antibodies to be associated with clinical protection.^{8,9} Based on this evidence, we interpreted our results accordingly. We now acknowledge this limitation in our discussion section. For illustration, we present now the correlation (Spearman coefficient) between the variant-specific RBD antibodies IgG and variant-specific nAb IgG (against Wuhan, BA1.1, BA4/5 and BQ1.1) in a subset of participants (n=89) from Timepoint 11 in whom we measured nAb for another project, showing correlation coefficients to a minimum of 0.7. These correlations have been added to the supplementary material (Supplementary eFigure 1).

We evaluated the correlation between anti-RBD antibodies for Wuhan, BA.1.1, BA.4/5, and BQ1.1 with their respective neutralizing antibodies (nAb) measured in a subset of individuals at T11.

^{26,27} The Spearman correlation coefficient between anti-RBD antibodies and nAb ranged from 0.70 to 0.78 (Supplementary eFigure 1).

Third, we did not evaluate neutralizing antibodies, although there is a high correlation between anti-RBD IgG levels and neutralizing antibodies^{42,43} with a r_s range between 0.70 and 0.78.

7. Correlation analysis in Supp Fig 2. The Pearson's correlation test was applied. However, there is no information whether the data is normally distributed or parametric. If this is not the case, then Spearman's correlation should be applied. Please provide the details.

ANSWER: We evaluated the distribution of each antibody MFI, log-transformed, and all of them have a fairly normality distribution, as per plots below. We did not conduct any null-hypothesis testing because of their high rate of false positives as the sample size increases. We clarified it in the methods.

T10

T11

the seven RBD from variants, and T10 and T11 with the Pearson correlation coefficient, after -evaluating their normal distribution graphically.

8. Figure 1 and Supp Fig. 1 should be combined for better understanding.

ANSWER: We agree and combine previous Figure 1 and Supp Fig. 1, which is now our current Figure 1.

Figure 1. CovidCatCentral Cohort history

Panel A. *Number of individuals in each of the CovidCatCentral cohort timepoints for the first-vaccinated and first-infected groups before study inclusion selection. The blue numbers represent the first-vaccinated, red numbers represent the first-infected, and the orange numbers show the total number of individuals at each time point. Green numbers indicate new individuals entering the study, grey numbers indicate individuals from any previous timepoints but not consecutive, and black numbers indicate individuals who followed up from one consecutive timepoint to the next.*

Panel B. *Prevalence of the different COVID-19 variants circulating in Spain from 2020 to 2023 and related to the timepoints.*

Panel C. *Temporal distribution of infections and vaccinations stratified by first exposure type among the analyzed population. Omicron infections started between T8 and T9.*

9. Lines 306-307: It is unclear about the sampling time point for the following, ‘and it was shorter for the first- vaccinated group compared to the first-infected group for T10 and similar for T11’.

ANSWER: The intervals mentioned in this sentence refer to the time (in days) from the last exposure (i.e., vaccination or infection) to the blood sampling used in the analysis. When we compared this interval for the first-vaccinated compared with the first-infected, it was shorter for the first-vaccinated (i.e., the time from blood sampling to the last exposure was shorter) in both T10 (152 [21-225] x 189 [44-319] days) and T11 (211 [162-382] x 227 [165-379] days), Table 1. We tried to clarify the sentence.

The time from the last exposure (i.e., last vaccine or infection) relative to the blood sampling used in the analysis was 173 [27-299] days for T10 and 224 [164-380] days for T11; and it was shorter for the first-vaccinated group compared to the first-infected group for T10 and similar for T11 (Table 2).

10. Figure legends are missing key information to be interpretable.

ANSWER: We apologize for this and in the revised version, we have improved the footnotes, making the figures clearer.

11. Figure 2 states it is a comparison of differences of antibody responses in the donors primed by infection and vaccination. It is unclear how this is represented in this data.

ANSWER: Thank you for the opportunity to clarify this point. The figure (now Figure 3) shows on the Y-axis the difference in antibody levels between those primed by vaccination minus those primed by infection (ie, it is the beta coefficient from the linear mixed model, after transformation in percent change). We clarified the reviewed Y-axis label and explained it better in the figure footnote.

Figure 3. Percent change on IgA and IgG antibody responses to RBD at T10 and T11 in first-vaccinated group minus first-infected group*

12. How was the assay they use validated? Especially the N specific assay which shows high titers for all donors at TP 5.

ANSWER: The assay used for anti-N was developed at our Lab at the beginning of the pandemic.^{1,2} It is also worth mentioning that while for anti Spike IgG antibodies we used the dilution of 1/5000, for anti-N we used 1/500, apologies that this information was missing in the original submission. We have now revised Figure 5 and clarified these points.

IgG, and additionally at 1:5000 for IgG to avoid saturated anti-S IgG levels. Anti-N IgG was tested at 1:500 dilution. - To quantify IgA, samples and controls were pretreated with anti-

The dashed gray lines represent the mean cutoff values for each antibody and isotype to be considered positive. We used the mean cutoff across the timepoints to provide a single line in the figure. Cutoff values: For IgA, the mean cutoff values were 2018.49 (SD = 530.17) for the N, 441.30 (SD = 136.65) for RBD, and 421.53 (SD = 60.68) for S. For IgG, the mean cutoff values were 28128.12 (SD = 9968.65) for N, 103.33 (SD = 28.58) for RBD, and 246.40 (SD = 99.78) for S. -Shaded areas represent 95% confidence intervals.

Minor

1. Please provide sequence details of AA/number positions for S, RBD and N antigens used in this study. Further, it would be great to include their QC data e.g. SDS-PAGE and Native-PAGE gel images.

ANSWER: We have now added them to the manuscript.

N Wuhan	full-length	1-416	
S Wuhan	aminoacids	1-1213	MFIF...IKWP
RBD Wuhan	aminoacids	319-541	RVQP...CVNF

Regarding SDS-PAGE and Native-PAGE gel images, we think it is not necessary to show them in the manuscript. These antigens have been already used in several studies and publications and have shown a high performance (see validation of assay references above).

We measured IgA and IgG plasma levels (median fluorescence intensity, MFI) to the full-length (FL) SARS-CoV-2 nucleocapsid (N) (amino acids 1-416); ~~and~~ Spike (S) antigens (amino acids 1-1213, amino acid sequence starting MFIF and ending IKWP), and the receptor binding domain (RBD) that lies within the S1 region (RBD ancestral, amino acids 319-541,

amino acid sequence starting RVQP and ending CVNF; and variants Delta, BA.1, BA.2, BA.4/5, BQ.1.1, and XBB), by quantitative suspension array technology assays (xMAP, Luminex), following a previously described protocol.^{21–23}

2. Supp Table 1: Table is incomplete on the right side for IgA response.

ANSWER: Thank you for highlighting this. We believe the PDF conversion cut out the end of the table. We now guarantee that the full table is readable.

ANSWER: Thank you for the attention to our work.

Reviewer #2 (Remarks to the Author):

ANSWER: Thank you for the attention to our work.

Reviewer #3 (Remarks to the Author):

In their manuscript, Ranzani and colleagues analyze antibody responses to different parts of SARS-CoV-2 and its variant over time and multiple exposures. The work is well done and the manuscript is interesting. But there are some points that need the authors' attention.

ANSWER: Thank you for your comments and appreciation.

Major points

1) Parts of the manuscript are a little hard to follow. The authors should make an effort to make it easier accessible for readers.

ANSWER: Thank you for highlighting this point. We carefully reviewed the manuscript, aiming to improve its readability.

2) While the work adds to our knowledge of the evolution of immune responses to SARS-CoV-2 over time, it is descriptive in nature and does not provide mechanistic insights.

ANSWER: We appreciate the reviewer's comment. While the study is largely descriptive, we provide mechanistic insights by analyzing cellular responses and the evaluation of number of previous exposure interactions and sequence. We also evaluate multiple Omicron RBD subvariants.

Importantly, our findings on how the initial exposure—whether by infection or vaccination—shapes subsequent immune imprinting offer a mechanistic framework to understand long-term immune evolution.

3) It has been reported that ratios between RBD and spike antibodies change over time. The authors should analyze this (since the titers have already been measured anyways).

ANSWER: Thank you for this suggestion. We now evaluated the effect of first exposure on the ratio between anti-RBDs and anti-S.

In this new figure added to the Supplemental material (eFigure 4), we observe a similar association of the IgG and IgA RBD/S ratio (higher in first vaccinated) as the association observed in the main analysis for higher anti-RBD response for those first-vaccinated compared with the first-infected.

We further evaluated the same models using the anti-RBD/anti-S antibody ratio as the outcome as potential better surrogate of nAb.

Overall, the results analyzing the anti-RBD/anti-S ratio showed similar pattern to the main analysis, except for higher uncertainty for IgA/anti-S ratio and anti-RBD XBB.1/anti-S ratio (Supplementary eFigure 4).

Minor points

1) Many abbreviations, including in the abstract are not defined. E.g. COVID-19, RBD, SARS-CoV-2 in the abstract. Main body: COVID-19, IRB, IPTG, SDS-PAGE, CHO, SD,

ANSWER: We expanded the first appearance of the acronyms.

2) Line 75-76: Is this the fact? There are several papers showing much higher peak antibody responses after initial vaccination than after initial infection. Please check the literature and add references to support that statement.

ANSWER: We agree that some studies reported the opposite, ie, higher peak after vaccination compared with infection. For instance, a recent study analysing anti-RBD antibodies, found much higher peak post-vaccination than post-(mild)-infection, but compared two doses vs. a single mild infection.¹¹ But overall the statement holds true when pooled the literature available for first exposure, such as the following systematic review and meta-analysis pooling 44 studies.¹⁰ This meta-analysis combined different types of antibodies, which might not be ideal for comparison, however could track time post-exposure and number of vaccine doses, focusing on the first-exposure. We revised the statement to focus on the systematic review and meta-analysis findings.

Regarding antibody dynamics, a systematic review and meta-analysis showed a higher antibody titer peak for infection-acquired antibodies compared to vaccine-induced antibodies, particularly for the first weeks post-exposure and after 6-months.⁹ In contrast, some studies showed the opposite, with higher peak of anti-RBD antibodies for first-vaccinated compared with first-infected.¹⁰

3) Line 80, line 249: It should be 'prime-exposure'.

ANSWER: Thanks. We corrected it.

4) Line 121: 'which left', instead of 'remaining'.

ANSWER: Thanks. We corrected it.

5) Line 131, line 260, line 275, line 347: Please do not use capitalized letters to start words mid-sentence. It should be 'spike', receptor binding domain' etc.

ANSWER: Thanks. We corrected 'spike,' 'receptor binding domain,' 'bivalent,' and similar terms where applicable. We retained capitalization for 'Cox' and 'Wilcoxon' as they are proper names.

6) Line 141: 'into a'

ANSWER: Thanks. We corrected it.

7) Line 157: 'batch'

ANSWER: Thanks. We corrected it.

8) Line 179: Please specify the actual penicillin and streptomycin concentrations.

ANSWER: We have added the information to the revised manuscript: 1% penicillin/streptomycin (10.000 U/ml, Thermo Fisher Scientific).

2.5x10⁵ thawed PBMCs were added to wells containing the stimulus (1 µg/mL of peptide concentration) or the negative control (only culture medium [TexMACS Medium (Miltenyi)- 1% penicillin/streptomycin (10.000 U/ml, Thermo Fisher Scientific)],

9) Line 248: 'effecy'?

ANSWER: We are sorry for the typos. We correct it to "effect".

10) Line 288: 'vaccinated'

ANSWER: We correct it to "vaccinated".

11) Line 354: 'infections'

ANSWER: We correct it to "infections".

12) 'et al.' should always be in italics.

ANSWER: We corrected those.

Reviewer #4 (Remarks to the Author):

This manuscript tried to addresses a critical public health topic of the antibody and cellular responses by investigating the effects of different initial exposure routes on immune responses. The manuscript presents interesting findings on the immune response dynamics based on the first exposure route to SARS-CoV-2 antigens. However, the significant methodological and interpretative issues undermine the reliability of the conclusions. The authors are encouraged to

address the following issues comprehensively to improve the rigor and reliability of the manuscript.

ANSWER: Thank you for your comments.

Study Design and Sample Limitations

1. Sample Size and Distribution Imbalance

1) 、 The sample sizes of the first-infected group (n=197) and the first-vaccinated group (n=160) are unbalanced, potentially affecting statistical power.

ANSWER: We thank the reviewer for this observation. While there is a small difference in sample size between the groups, we respectfully note that the statistical approach used—a linear mixed-effects model—is well-suited to handle unbalanced data structures of this kind. These models are robust to moderate sample size differences and are specifically designed to account for within-subject variability and unequal group sizes without introducing bias in fixed-effect estimates.

In our case, both groups are adequately powered individually ($n > 150$), and the difference in size (197 vs. 160) is relatively small and unlikely to meaningfully affect model stability or inference.

2) 、 The cellular assay is based on only 49 individuals with hybrid immunity, which is insufficient for robust subgroup comparisons (first-infected vs. first-vaccinated).

ANSWER: We acknowledge the limitation regarding the sample size of the cellular assay. Indeed, T-cell assays are technically more complex, time-consuming, and resource-intensive compared to antibody assays, which limits the number of samples that can be processed. Despite this constraint, the cohort of 49 individuals with hybrid immunity still allowed us to observe consistent patterns across key comparisons, including first-infected versus first-vaccinated individuals. Importantly, these data complement and reinforce the findings observed in the broader antibody dataset. Furthermore, the conclusions drawn from subgroup comparisons are interpreted with appropriate caution. Future studies with larger cohorts will be instrumental to confirm and expand upon these findings.

3) 、 Please provide a justification for the current sample size and its adequacy, or expand the sample size, particularly for highly exposed groups, to ensure statistical reliability.

ANSWER: We appreciate the reviewer's concern regarding sample size adequacy. The sample size used in our study reflects the total number of eligible individuals available within the study period and the inclusion criteria. Importantly, the statistical models applied —particularly the linear mixed-effects models— are well-suited for moderately sized cohorts and are capable of producing reliable estimates even with varying group sizes.

2. Timeframe and Virus Strain Variability

1) 、 The two groups had their first exposure in 2020 and 2021, involving different viral strains (Wuhan vs. Alpha). This undermines the comparability of the groups, as different strains may

trigger distinct immune responses. The differences between the Wuhan and Alpha strains likely contributed to variations in antibody responses, further complicating the comparison.

ANSWER: Thank you for the opportunity to discuss this topic. In the context of Spain's vaccine rollout and epidemic waves, these differences are not easily avoidable and, in fact, reflect the real-world sequence of exposures. Specifically, individuals in the first-infection group were primarily infected in 2020–2021, during waves dominated by pre-Omicron variants (predominantly with Wuhan, but four individuals with Alpha). In contrast, individuals in the first-vaccinated group were largely infected post-vaccination, when Omicron and its subvariants predominated. Given the marked antigenic differences between Omicron and earlier variants, the first infecting strain likely acts as an effect modifier in the effect of first-exposure type on antibody levels. This is an inherent aspect of our comparison and contributes to the observed differences. We also have adjusted the model by the number of probable VoCs (MS3 and MS4), and there are no major differences from the main analysis. We added this important topic in the discussion.

We ran four sensitivity analyses: MS1, using M5 (main model)4 but expanding the number of chronic comorbidities as three binary factors (cardio-metabolic, immunosuppressed and previous allergy); MS2, running M54 (main model) in those individuals without any history of asymptomatic infection; MS3, using M5 (main model) but instead of non-bivalent/bivalent, using number of previous mRNA and adeno-virus based vaccines; and MS4, running MS3 but instead of number of non-omicron/omicron, using number of previous infections by each variant. We further evaluated the same models using the anti-RBD/anti-S antibody ratio as the outcome as potential better surrogate of nAb. A summary of the models adjustment is on Supplementary eTable 1.

2) 、 The intervals between the last exposure and sampling (T10 vs. T11) differ between the groups, potentially affecting the observed antibody decay rates.

ANSWER: Yes, this is the reason we adjusted for this time interval in Model 5 and the sensitivity analyses. We adjusted for the time interval from the last infection and the time interval from the last vaccine.

3) 、 Please verify the data in Appendix Figure 1 against the main text, with particular attention to the timeframe and sample size information.

ANSWER: Thank you for the opportunity to clarify this. The timeline shows all participants' dynamics across timepoints, not necessarily fulfilling our inclusion criteria (we can see the totals 392 in T10 and 346 in T11). In the current analysis, after inclusion/exclusion criteria, we have 357 individuals (197 in first-infected and 160 in first-vaccinated): 289 participated in both T10 and T11, 48 only in T10 and 20 only in T11. This is the reason the numbers of our analysis are not a full match with size shown in the timeline. We clarified this in the footnote of the revised figure (currently our Figure 1).

Panel A. Number of individuals in each of the CovidCatCentral cohort timepoints for the first-vaccinated and first-infected groups before study inclusion selection. The blue numbers represent the first-vaccinated, red numbers represent the first-infected, and the orange numbers show the total number of individuals at each time point. Green numbers indicate new individuals entering the study, grey numbers indicate individuals from any previous timepoints but not consecutive, and black numbers indicate individuals who followed up from one consecutive timepoint to the next.

3. Quantification of Exposure History

1) 、 **The methods do not clearly explain how the sequence of exposures (e.g., infection followed by vaccination) was quantified or incorporated into the analysis.**

ANSWER: Please, see the answer below.

2) 、 **Please provide a detailed explanation of how exposure sequences were quantified and their influence on immune responses.**

ANSWER: The follow-up of the health-care workers cohort enabled us to comprehensively track all SARS-CoV-2 exposures throughout the study period. At each timepoint, detailed information on vaccine administrations as well as confirmed infections, detected by positive rapid antigen tests or PCR, were systematically collected. Additionally, undiagnosed infections were identified through our serology assays defined as a ≥ 4 -fold increase in anti-N antibody levels between timepoints in vaccinated individuals, or in anti-S and/or anti-N levels in non-vaccinated individuals. This approach allowed us to accurately capture the timing and order of both immunization and infection events for each participant.

In the models, we adjusted by the number of previous vaccines and the number of previous infections, separating them by whether the previous infections were by Omicron or not, and whether the vaccines were bivalent or not. In the revised version, we expanded it, adjusting also by type of vaccine and VoC.

To account for the specific sequence of previous exposures, we estimated the percent change in anti-RBD antibody levels between first-exposure groups for histories where we had meaningful comparison (ie, same sequence differing only in the first-exposure) and enough sample size, as those shown in the whole sequence history of exposures in Figure 5. There we could evaluate 8 sequences in four pairs that the only difference was the first exposure.

We clarified it in the methods.

follows: M0, without adjustment; M1, M0 plus age (restricted cubic spline with 3df) and sex; M2, M1 plus number of chronic comorbidities and tobacco smoking status; M3, M2 plus number of non-Omicron and Omicron symptomatic infections (as factors), and number of non-Omicron and Omicron asymptomatic infections (as factors); M4, M3 plus number of non-bivalent and bivalent vaccines (as factors); and finally, M5 (main model), M4 plus days from last infection (restricted cubic spline with 3df) and days from last vaccine (restricted cubic spline with 3df).

We ran four sensitivity analyses: MS1, using M5 (main model) but expanding the number of chronic comorbidities as three binary factors (cardio-metabolic, immunosuppressed and previous allergy); MS2, running M5 (main model) in those individuals without any history of asymptomatic infection; MS3, using M5 (main model) but instead of non-bivalent/bivalent, using number of previous mRNA and adeno-virus based vaccines; and MS4, running MS3 but instead of number of non-omicron/omicron, using number of previous infections by each variant.

A summary of the models adjustment is on **Supplementary eTable 1.**

To evaluate whether the difference between first-infected and first-vaccinated occurs mainly due to specific previous vaccination and infection histories, ie.i.e., the exact sequence of exposures, we fitted a model adjusted as M5 but using the exact sequence of exposures instead of number of previous infections and vaccinations. To create this sequence, we considered asymptomatic and symptomatic infections as equal previous infections and non-bivalent and bivalent vaccines as equal previous vaccines, to deal with sparse data and improve power. We fitted the model with a categorical variable containing the 9 most

common history combinations (i.e., combinations with at least 20 samples, n=212 individuals), while collapsing those below 20 samples as “others” (n=145 individuals). Among the 9 most common history combinations, we chose those with a meaningful contrast to understand the first exposure effect (i.e., those with the same number of exposures, but differing on the prime-exposure), resulting in eight sequences and four comparisons.

Statistical Analysis Issues

1. Complexity and Potential Bias in Statistical Models:

1) The main model (M5) incorporates a large number of confounding variables, raising concerns about overfitting and multicollinearity (e.g., between vaccine doses and time intervals).

ANSWER: Thank you for the opportunity to clarify this point. Our modeling strategy aimed to adjust for important confounders to minimize bias in estimating antibody response, based on the conceptual framework of confounding adjustment (and not on p-values or automatic selection).¹² While the model includes multiple covariates, overfitting is unlikely given the structure and size of the dataset (n=646 samples), especially in the context of linear mixed-effects models, which are designed to handle correlated repeated measures and moderate to large covariate sets (in contrast with generalized mixed models as binomial).^{13,14}

To address the concern about multicollinearity, we evaluated variance inflation factors (VIFs) among the fixed effects. None exceeded conventional thresholds (e.g., VIF >5), except the spline terms as expected, suggesting that multicollinearity is not a significant issue in our specification. Furthermore, we verified model robustness by conducting sensitivity analyses excluding or combining potentially collinear terms (e.g., dose number and time since last dose), and results remained consistent.

confidence intervals (CI) for the estimates from the model. All linear mixed models were evaluated regarding residual diagnostics and multicollinearity with variance inflation factors (VIF). All models were within the expected assumptions and there was no evidence for multicollinearity (i.e., VIF<5).

2) High exposure counts are only observed in the first-infected group, creating an imbalance for interaction effect analysis.

ANSWER: Thank you for the opportunity to clarify this point. The reviewer is correct, for this reason, the interaction analysis was restricted to individuals who had between 3 and 6 exposure counts. We further clarified this issue.

exposures (as factor). We evaluated a potential statistical interaction with a likelihood ratio

test comparing the model with and without the interaction term. This analysis was

conducted among those with a minimum of 3 and maximum of 6 previous exposures,

because only those in the first-infected group had more than 6 previous exposures (n=3

individuals).

3) Simplify the statistical model by reducing unnecessary variables. Please provide diagnostic results (e.g., AIC/BIC, residual analysis) to validate model quality.

ANSWER: As discussed earlier, we did not use automatic variable selection methods to adjust for confounding, in line with established guidance in the literature that discourages such approaches for association/causal inference, as they are more appropriate for prediction modeling.

That said, we conducted model diagnostics,, including residual distribution assessments and fit statistics to evaluate model quality. These diagnostics did not reveal violations of model assumptions or poor model performance.

To further address the reviewer’s point, we ran exploratory models using stepwise selection based on F-statistics, starting from the full set of 29 variables included in models M1–M5 and sensitivity analyses MS1–MS4. Across the 14 resulting models:

- The first-exposure effect was retained in 7 out of 7 IgG models and 5 out of 7 IgA models.
- In the two IgA models where it was excluded (anti-RBD Wuhan and anti-RBD Delta), it was dropped only at the 25th and 26th elimination steps, with only 2 variables remaining — indicating it was among the last to be removed.

These results (not included in the manuscript) confirm that the first-exposure effect remains robust.

Covariates selected in at least 50% of models (ie, appeared at least 7 times over 14) in an exploratory analysis

	All (n=14)	IgA (n=7)	IgG (n=7)
Interval from last vaccin	13	7	6

First-exposure	12	5	7
Number of asymptomatic omicron infections	12	5	7
Number BA.1	12	5	7
Number BA.2	12	5	7
Number BA.4/5	12	5	7
Number BQ.1.1	12	5	7
Number XBB	12	5	7
Number symptomatic omicron infections	12	5	7
Interval from last infection	10	3	7
Age	7	0	7

2. Conservative Correction for Multiple Comparisons

1) **Bonferroni correction, while controlling type I error, is overly conservative and risks type II errors (false negatives).**

ANSWER: The reviewer is correct. We now have provided the p-values comparison for unadjusted, and corrected by Bonferroni and FDR methods, as indicated in the response below.

2) **Present results using both Bonferroni and False Discovery Rate (FDR) corrections, and compare unadjusted and adjusted p-values to evaluate their impact.**

ANSWER: Thank you for your suggestion. We now provide a figure in the supplement (Supplementary eFigure 4) with unadjusted and adjusted p-values for Bonferroni and FDR.

We did not observe any major changes in the interpretation of the results if we base them on p-value, except for three comparisons when Bonferroni was close to $p=0.05$ and FDR was below 0.05 (two contrasts for IgA and anti-RBD Delta and one contrast for anti-RBD BQ1.1, Table below). So, although it might have been a bit conservative on the confidence intervals, we did not expect any main change in our interpretation. We have added this to the manuscript.

	Isotype	Analyte	Unadjusted	Bonferroni	FDR
VacVacInf vs InfVacInf	IgA	anti-RBD Delta	p=0.0135	p=0.0539	p=0.0270
VacVacVaccInf vs InfVacVacVac	IgA	anti-RBD Delta	p=0.0130	p=0.0520	p=0.0270
VacVacVaccInf vs InfVacVacVac	IgA	anti-RBD BQ1.1	p=0.0215	p=0.0858	p=0.0429

In a sensitivity analysis, we evaluate these comparisons without multiplicity correction and using false-discovery rate (FDR) method.

*When we explored FDR for multiplicity correction, we did not observe any major changes in the interpretation of the results analysing p-values <0.05, except for three comparisons when Bonferroni was close to p=0.05 and FDR was below 0.05 (two contrasts for IgA and anti-RBD Delta and one contrast for anti-RBD BQ1.1, **Supplementary eFigure 5**).*

3. Fitting of Antibody Dynamics

1) The LOWESS curve used for antibody dynamics is prone to edge effects, potentially leading to inaccuracies at the endpoints.

ANSWER: The reviewer is correct regarding potential edge effects in LOWESS smoothing. To address this concern, we carefully looked at the fit of the curve, especially near the endpoints, and decreased our span. We confirm that the smoothed trends remain consistent with the observed data, and no artificial inflection points were introduced at the boundaries.

2) Provide details on LOWESS parameters and perform sensitivity analyses to verify the robustness of the fits. Consider alternative models, such as generalized additive models (GAM), for better reliability.

ANSWER: For the LOWESS, we used the default parameters from “geom_smooth” from *ggplot2* package (degree: “2 - quadratic”; family: “gaussian”; surface: “interpolate”) and adapted the span from 0.75 to 0.50 (it is, having less flexible the degree of smoothing). In order to provide a less subjective decision, we also modelled it using GAM, with the *mgcv* package, and let its robust algorithm¹⁵ automatically set the non-linear thin-plate spline, as suggested. The results of both figures are comparable, as shown below.

Figure 6. Longitudinal Wuhan antibody levels against nucleocapsid (N), receptor-binding domain (RBD), and spike (S) proteins overtime (modelled with LOWESS)

The blue and red solid lines represent the fitted curve calculated using the LOWESS (locally estimated scatterplot smoothing, *ggplot2::geom_smooth*) method with the following parameters: degree: “2 - quadratic”; family: “gaussian”; surface: “interpolate”, span 0.50. The dashed gray lines represent the mean cutoff values for each antibody and isotype to be considered positive. We used the mean cutoff across the timepoints to provide a single line in the figure. Cutoff values: For IgA, the mean cutoff values were 2018.49 (SD = 530.17) for Ne, 441.30 (SD = 136.65) for RBD, and 421.53 (SD = 60.68) for S. For IgG, the mean cutoff values were 28128.12 (SD = 9968.65) for N_FI, 103.33 (SD = 28.58) for RBD, and 246.40 (SD = 99.78) for S. Shaded areas represent 95% confidence intervals.

eFigure 5. Longitudinal Wuhan antibody levels against nucleocapsid (N), receptor-binding domain (RBD), and spike (S) proteins overtime (modelled with GAM)

The blue and red solid lines represent the fitted curve calculated using a generalized additive model (GAM), with thin-plate penalized splines and the spline parameters were determined by the algorithm available on the R package *mgcv*. The dashed gray lines represent the mean cutoff values for each antibody and isotype. We used the mean cutoff across the timepoints to provide a single line in the figure. Cutoff values: For IgA, the mean cutoff values were 2018.49 (SD = 530.17) for N, 441.30 (SD = 136.65) for RBD, and 421.53 (SD = 60.68) for S. For IgG, the mean cutoff values were 28128.12 (SD = 9968.65) for N FI, 103.33 (SD = 28.58) for RBD, and 246.40 (SD = 99.78) for S.

*We plotted the MFI levels against each timepoint and fitted a locally weighted scatterplot smoothing (LOWESS) curve. The LOWESS was fit with the `ggplot2::geom_smooth` function with the following parameters: `degree: "2 - quadratic"`; `family: "gaussian"`; `surface: "interpolate"`, `span 0.50`. To evaluate another method to account for the nonlinearity, we modelled it using a generalized additive model (GAM), with the *mgcv* package, and let its robust algorithm³¹ automatically set the non-linear thin-plate spline.*

4.p-value

1) The manuscript applies different p-value thresholds for main effects (≤ 0.05) and interaction terms (≤ 0.10). Could the authors clarify the rationale behind using a less stringent threshold for

the interaction term and discuss the potential implications of this choice on the interpretation of the results?

ANSWER: Thank you for the opportunity to clarify this point. The choice of a less stringent threshold for the interaction term is because the evaluation of interaction coefficients is usually underpowered and it is common practice to use less stringent thresholds. We did not expect any direct implication in the interpretation of the results, which was largely not based on the dichotomy of statistical significance.

Results and Interpretation

1.Lack of Dynamic Analysis:

1) The study only compares T10 and T11 timepoints, missing insights into the dynamic trends of immune responses over time.

ANSWER: Unfortunately, we do not have RBD antibody data against Omicron subvariants, the main objective of the current study, beyond T10 and T11. For this reason, we showed the longitudinal data of RBD, S, N against Wuhan over time for which we have both first-infected and first-vaccinated groups (from T5 to T11).

To explore the observed difference between first-vaccinated and first-infected, we evaluated antibody kinetics for IgA and IgG against Wuhan with anti-S, anti-N and anti-RBD antibodies using the data available from T5 to T11.

2) Incorporate longitudinal analyses to better characterize antibody decay rates and T-cell dynamics.

ANSWER: Unfortunately, we do not have these data for more timepoints than T10 and T11 for Omicron anti-RBDs.

2.Inadequate Adjustment for Confounding Factors:

1) Significant group differences exist in age ($p = 0.006$) and comorbidities (e.g., cardio-metabolic conditions, $p = 0.016$, and allergy history, $p = 0.024$), which may confound immune response comparisons.

ANSWER: The reviewer is correct. And our models are adjusted by these variables. Age is adjusted for since Model 1, and comorbidities since Model 3, with a sensitivity analysis expanding to account for the specific cited comorbidities (MS1). Please, see the new eTable 1 with the full model specifications.

2) Either adjust for these confounders in statistical models or use propensity score matching to balance the groups.

ANSWER: We are sorry for not being clear, but they are adjusted in the linear mixed models. We clarified this in the manuscript.

3. Unclear Definitions of Asymptomatic Infections:

1) Asymptomatic infections are inferred solely based on antibody level changes, which could be influenced by other factors (e.g., vaccination, individual immune traits), leading to potential misclassification.

ANSWER: We acknowledge the reviewer's concern regarding potential misclassification of asymptomatic infections based solely on antibody level changes. To address this, we incorporated both established serological criteria and antigen-specific responses to distinguish infection-induced antibody increases from those driven by vaccination or individual immune variability.

Specifically, we selected the 4-FC threshold because it is widely recognized in the scientific literature as an indicator of seroconversion following infection or vaccination. Both the World Health Organization (WHO) and the European Medicines Agency (EMA) explicitly reference a 4-fold rise in antibody titers as a criterion for diagnosing infection or evaluating vaccine-induced responses. WHO guidelines, for instance, state that "seroconversion or a 4-fold increase in IgG-specific antibody titers are suitable for the diagnosis of COVID-19" (Long et al., *Nat Med*, 2020; <https://doi.org/10.1038/s41591-020-0897-1>). EMA guidelines similarly note the use of a ≥ 4 -fold rise in antibody titers as a standard metric for assessing seroconversion following vaccination (EMA/CHMP/VWP/164653/05 Rev1, 2018; https://www.ema.europa.eu/en/documents/scientific-guideline/draft-guideline-clinical-evaluation-vaccines-revision-1_en.pdf). This threshold has been adopted in numerous studies of COVID-19 (e.g., Trieu et al., *J Infect Dis*, 2021; Zhu et al., 2020, *Lancet*; Xia et al., 2021, *Lancet*; Dobaño et al., 2021, *BMC Med*) and in serological assessments of other infectious diseases, including Shigella (Chisenga et al., *PLoS One*, 2021), Influenza (e.g., Hsu et al., *BMC Infect Dis*, 2014; Chang et al., *Vaccines*, 2021), Pertussis (Ladhani et al., *Clin Infect Dis*, 2015), Polio (Niang et al., *Vaccine*, 2020), and Dengue (Durbin et al., *J Infect Dis*, 2016). Given this broad precedent, we considered the 4-fold threshold an appropriate and conservative starting point for detecting potential SARS-CoV-2 infections in our study.

Importantly, to reduce the likelihood of misclassifying vaccine-induced responses as infections, we incorporated nucleocapsid (N)-specific antibody measurements, which are not expected to rise following mRNA vaccination. In individuals who received a vaccine between two study timepoints, an infection was only inferred if the fold change in IgG or IgA against the N antigen exceeded 4. We do not expect any individual immune traits to raise the antigen-specific antibody response rather than a natural infection.

2) Use additional evidence (e.g., PCR results) to corroborate the classification of asymptomatic infections.

ANSWER: We appreciate the reviewer's concern regarding asymptomatic case detection. Unfortunately, our cohort did not include participants who underwent active routine PCR testing, making direct validation with PCR results not possible. However, to ensure the robustness of our antibody-based classification, we evaluated the proportion of confirmed diagnosed infections (ie,

those with a positive RT-PCR or rapid antigen test) detected using our method based on fold-change (FC) increases in antibody levels. Using our approach, we were able to identify: 6/7 (85%) infections at T6-T7, 5/5 (100%) at T7-T8, 39/42 (93%) at T8-T9, 68/89 (76%) at T9-T10, and 29/33 (88%) at T10-T11, supporting the validity of our antibody-based classification.

intervals. This reduction in fold-change increase is attributed to elevated antibody levels at later timepoints. Using this approach, we were able to identify infections when compared to RT-PCR/RDT test as follow: we detected 6/7 (85%) infections at T6-T7, 5/5 (100%) at T7-T8 39/42 (93%), at T8-T9, 68/89 (76%), at T9-T10, and 29/33 (88%) at T10-T11, supporting the validity of our antibody-based classification.

Minor points

1. In the formula " $[10^{\beta} - 1] * 100$ ", " β " should use the correct Greek symbol with proper spacing.

ANSWER: Done. Thank you.

2. "effecy" is a spelling error and should be corrected to "effect."

ANSWER: We are sorry for the typos. We corrected it.

3. The term "non bivalent" should be written as "non-bivalent" consistently throughout the manuscript.

ANSWER: We corrected it.

4. When LOWESS is introduced, its full name ("Locally Weighted Scatterplot Smoothing") should be provided.

ANSWER: Thank you, we expanded it at its first appearance.

5. Inconsistent formatting of terms like "anti-RBDWuhan" should be corrected to "anti-RBD Wuhan."

ANSWER: The reviewer is correct. We reviewed the manuscript to ensure consistency throughout the manuscript.

6. "...with individuals being followed-up up to the starting point...", the repeated "up" should be corrected by removing one instance.

ANSWER: Thank you, we corrected it.

REFERENCES

1. Dobaño, C. *et al.* Immunogenicity and crossreactivity of antibodies to the nucleocapsid protein of SARS-CoV-2: utility and limitations in seroprevalence and immunity studies. *Translational Research* **232**, 60–74 (2021).
2. Dobaño, C. *et al.* Highly Sensitive and Specific Multiplex Antibody Assays To Quantify Immunoglobulins M, A, and G against SARS-CoV-2 Antigens. *J Clin Microbiol* **59**, e01731-20 (2021).
3. Wajnberg, A. *et al.* Robust neutralizing antibodies to SARS-CoV-2 infection persist for months. *Science* **370**, 1227–1230 (2020).
4. Servellita, V. *et al.* Neutralizing immunity in vaccine breakthrough infections from the SARS-CoV-2 Omicron and Delta variants. *Cell* **185**, 1539-1548.e5 (2022).
5. Rockstroh, A. *et al.* Correlation of humoral immune responses to different SARS-CoV-2 antigens with virus neutralizing antibodies and symptomatic severity in a German COVID-19 cohort. *Emerging Microbes & Infections* **10**, 774–781 (2021).
6. Guiomar, R. *et al.* Monitoring of SARS-CoV-2 Specific Antibodies after Vaccination. *Vaccines* **10**, 154 (2022).
7. Aguilar, R. *et al.* RBD-Based ELISA and Luminex Predict Anti-SARS-CoV-2 Surrogate-Neutralizing Activity in Two Longitudinal Cohorts of German and Spanish Health Care Workers. *Microbiol Spectr* **11**, e0316522 (2023).
8. Martín Pérez, C. *et al.* Correlates of protection and determinants of SARS-CoV-2 breakthrough infections 1 year after third dose vaccination. *BMC Med* **22**, 103 (2024).
9. Pérez, C. M. *et al.* Determinants of Antibody Levels and Protection against Omicron BQ.1/XBB Breakthrough Infection. Preprint at <https://doi.org/10.1101/2024.10.11.24315296> (2024).
10. Zhang, Q. *et al.* COVID-19 antibody responses in individuals with natural immunity and with vaccination-induced immunity: a systematic review and meta-analysis. *Syst Rev* **13**, 189 (2024).

11. Harrache, A. *et al.* Anti-RBD IgG dynamics following infection or vaccination. *Vaccine* **42**, 126464 (2024).
12. Lederer, D. J. *et al.* Control of Confounding and Reporting of Results in Causal Inference Studies. Guidance for Authors from Editors of Respiratory, Sleep, and Critical Care Journals. *Annals ATS* **16**, 22–28 (2019).
13. West, B. T., Welch, K. B. & Galecki, A. T. *Linear Mixed Models: A Practical Guide Using Statistical Software*. (CRC Press, Boca Raton, 2015).
14. Harrell, F. E. *Regression Modeling Strategies*. (Springer, Cham, 2015).
doi:10.1007/978-3-319-19425-7.
15. Wood, S. N. Fast Stable Restricted Maximum Likelihood and Marginal Likelihood Estimation of Semiparametric Generalized Linear Models. *J R Stat Soc Series B Stat Methodol* **73**, 3–36 (2011).

REVIEWERS

REVIEWER COMMENTS

Reviewer #1 (Remarks to the Author):

The current version of the manuscript is substantially improved. I am content with the changes made and have no further concerns.

ANSWER: Thank you for your comments.

Reviewer #4 (Remarks to the Author):

I appreciate the authors' thorough response to the first-round review comments and their revisions, and I commend the improvements made in the clarity and structure of the manuscript. This study addresses a timely and important question regarding how the sequence of SARS-CoV-2 exposures influences the evolution of immune responses. However, there remain some critical issues in the interpretation of results and statistical methods that require further clarification and strengthening. I encourage the authors to carefully consider the following suggestions to enhance the scientific rigor and persuasiveness of the manuscript. The detailed review comments are as follows.

ANSWER: Thank you for your comments.

1. The subgroup of 49 individuals used for T-cell analysis appears underpowered to support robust comparisons between primary infection and primary vaccination. Although this limitation is acknowledged in the discussion, it should also be clearly stated in the abstract and conclusion. Descriptions such as "consistent patterns" may overstate the certainty of the findings and should be avoided or qualified with appropriate language. Please also add relevant cautionary notes in figure legends and the methods section.

ANSWER: We agree with the reviewer and added a statement about the potential limited power for the T-cell analysis in the abstract, conclusion and footnote of Figure 7. Nevertheless, we are aware that this is not necessarily a small sample size considering T-cell analysis.

Abstract

In contrast to the humoral response, the T-cell response was higher in individuals first exposed through infection, although T-cell findings may be underpowered because of limited sample size.

Discussion

In contrast, T-cell responses against Wuhan antigens were of greater magnitude in those first-infected compared to those first-vaccinated; this conclusion is limited by the relatively small cellular assay sample size.

Conclusion

In contrast to humoral response, T-cell response against Wuhan was higher in those first exposed through infection, although T-cell findings may be underpowered because of limited sample size.

Figure 7 Footnote

Note: T-cell findings may be underpowered because of limited sample size.

2.If feasible, report statistical power or confidence intervals for subgroup analyses to help readers assess the reliability of the findings. If not possible, a brief note discussing the limitations of statistical power in the methods or discussion would be helpful.

ANSWER: We appreciate the reviewer's suggestion. We did not conduct a post-hoc power analysis, as such analyses are generally considered uninformative: once the data are collected, the observed confidence intervals already reflect the precision and statistical uncertainty of the estimates (doi: 10.1097/SLA.0000000000003296; 10.1002/gepi.22464). In line with current reporting standards, we present confidence intervals for all subgroup analyses to allow readers to directly assess the precision of the findings. In the revised discussion, we have also added a note explicitly acknowledging the limited statistical power of subgroup analyses and cautioning that these results should be interpreted as exploratory.

Discussion

It is worth mentioning that we may have been underpowered for some analyses of the humoral response, particularly those examining the interaction between the first-exposure group and the number of previous exposures as well as when evaluating complete exposure history. Although we a priori set the type I error for the interaction test to 10% to increase sensitivity, simulation studies showed that this does not increase power to detect interactions;⁴⁶ thus these analyses should therefore be considered exploratory.

3.The modeling framework remains insufficiently detailed. Although the authors provided a table summarizing covariate adjustments across different models, it is unclear whether interaction terms were included or evaluated. Please clarify if any interaction analyses were performed. If not, the authors should consider conducting interaction analyses where appropriate, such as when there is biological rationale or prior evidence suggesting potential interactions.

ANSWER: The main models have no interactions, because we consider the unique biological rationale to support the interaction tested in a sensitivity analysis by number of previous exposures. We clarified it.

Methods

A summary of the model's adjustment is on **Supplementary eTable 1**. **In the previous models, we did not consider any interaction term.**

4.To address concerns about multicollinearity, consider including a summary of variance inflation factor (VIF) values, either in the main text or supplementary materials. It would also be helpful to share full regression outputs—including coefficients and p-values—so that readers can directly assess the results.

ANSWER: We agree with the reviewer and add the VIF information for all covariates, for models of IgA and IgG, and seven RBDs, from the main model (M5) to the supplementary material (now **eFigure 2**). Regarding outputs, we have all the information available for the reader in the main manuscript and supplementary material, showing the transformation of coefficients to %-change of the main exposure and covariates of interest (eTable 4). We do not rely on P-values for interpreting the results, as discussed.

Supplementary eFigure 2. Variance Inflation Factors (VIFs) from the main model (M5) for IgA and IgG and the seven RBDs.

The vertical dashed black lines are the VIF of 5, which might indicate multicollinearity. We plotted the spline terms for age, time from last infection and time from last vaccine, nevertheless the interpretation of VIF values for spline terms is not straightforward.

5. The manuscript mentions FDR correction, but it is unclear whether the primary findings remain statistically significant after adjustment. Please state this explicitly in the main text, and consider using visual indicators (e.g., color or symbols) in figures to highlight adjusted significance levels.

ANSWER: We used FDR as a sensitivity analysis for multiplicity correction as compared to the Bonferroni (our pre-specified correction method) following the reviewer suggestion during R1. We cited in the results text what had changed if considered FDR instead of Bonferroni and the figure showing their changes were shown in the supplementary material, with a line showing the 0.05 threshold. We now improved the text and added a symbol showing if the “0.05” statistical significance threshold changed with FDR.

Remembering the 95% CI of the figure 5 are already adjusted by Bonferroni, as stated in the footnote. Thank you for the opportunity to improve our work.

Figure 5. Percent change contrasting first-vaccinated group minus first-infected group in four pairs of sequences of previous exposures for IgA and IgG against RBD at T10 and T11*

* Estimates from linear mixed models estimating the %MFI increase in different exposure histories. The models accounted for repeated measurements in the same individual with a random intercept per individual. The model was adjusted adapting the main model (M5), i.e., excluding the number of previous infections and vaccines and using the history per se: age (restricted cubic spline with 3df) + sex + number of chronic comorbidities and tobacco smoking status + days from last infection (restricted cubic spline with 3df) + days from last vaccine (restricted cubic spline with 3df) and a factor with the history of the most common combinations. The error bars represent the 95% confidence interval corrected by multiple comparisons with the Bonferroni method. Green color represents confidence intervals that do not include zero. The symbols represent the contrasts that achieved the statistical threshold of $p \leq 0.05$ accordingly with multiplicity correction methods: * $p \leq 0.05$ without any correction; † $p \leq 0.05$ with false-discovery rate (FDR) correction, and ● $p \leq 0.05$ with Bonferroni correction.

6. The observed antibody trends are derived using LOWESS smoothing, but no quantitative or visual evidence is provided to rule out artifacts from boundary effects. A more systematic evaluation is needed to demonstrate that the observed trajectories are robust and not driven by the smoothing procedure itself.

We have performed a comprehensive set of sensitivity analyses to evaluate whether the observed antibody trajectories are artifacts of the LOESS smoothing procedure:

1 - Alternative LOESS specifications:

We refitted the smoothers using degree = 1 and family = "symmetric" (instead of degree = 2 and family = "gaussian", as in the original submission). These settings reduce sensitivity to local outliers and down-weight boundary influence. The resulting trajectories (Figure R1a) were consistent with those originally reported, confirming that the trends are not driven by the default LOESS parameterization.

2 - Span sensitivity:

We varied the smoothing span between 0.4–0.7 using both robust and non-robust LOESS (Figures R1b, c). Across this range, the qualitative trajectories remained stable.

3 - Cluster bootstrap by participant:

We performed a participant-level bootstrap, resampling individuals with replacement, refitting the LOESS smoother on each resample, and constructing percentile-based confidence bands (Figure R2a). The bootstrap 95% CI fully contained the originally reported trajectories, further supporting robustness to both smoothing and within-participant correlation.

4 - Boundary stress test:

To directly assess boundary effects, we refit the smoothing procedure excluding the first or last timepoint (T5 or T11). As shown in Figure R2b, the internal portions of the curves were virtually unchanged, demonstrating that the observed trends are not artifacts of the boundary points.

5 - Raw data summaries:

We calculated and plotted timepoint-specific medians and interquartile ranges of log₁₀ (MFI) for each group (Figure R2c). These summaries closely tracked the smoothed trends, demonstrating that the main features are already evident in the raw data without any smoothing assumptions.

6 - GAM models:

In the previous revision, we additionally reported trajectories estimated from generalized additive models with thin-plate splines and smoothing parameters selected by the mgcv algorithm. These GAM fits corroborated the LOESS-based findings.

Across all robustness checks, the qualitative differences between groups and the shape of the trajectories were preserved, indicating that the reported antibody dynamics are not artifacts of the smoothing procedure but robust features of the data.

Figure R1. Robustness of antibody trajectories to smoothing procedure. a. Robust LOESS (span = 0.5, degree = 1 (local linear fit), symmetric (robust to outliers)). **b.** Span sensitivity for LOESS (degree = 2, gaussian) (span: 0.4–0.7). **c.** Span sensitivity for LOESS (degree = 1, symmetric) (span: 0.4–0.7).

Figure R2. Robustness of antibody trajectories to smoothing procedure. **a.** Participant-level cluster bootstrap (300 iterations) CIs (95% pointwise percentiles). **b.** Boundary stress evaluation (re-fitting after dropping the first or last timepoint). **c.** Raw data summaries: median and interquartile range of log₁₀(MFI).

ANSWER: We added these panels to the supplement (now eFigure 7) as their methods description.

7.A relaxed significance threshold ($p \leq 0.10$) is used for interaction terms without prior specification or justification. While reduced power for interactions is a known issue, applying a looser threshold post hoc without a pre-defined rationale may introduce bias. Consider using a consistent threshold (e.g., $p \leq 0.05$) or providing a formal justification (e.g., through power analysis or pre-specified plans) for differential treatment.

ANSWER: We apologize for not being clear in the previous review. First, we clarify that the use of $p \leq 0.10$ for testing the interaction was pre-specified in our analysis plan, given the well-recognized low statistical power of interaction tests, especially in the context of our 2×4 categorical interaction. While we unfortunately do not have a publicly available statistical analysis plan, we emphasize that this decision was made a priori and not post hoc. Second, the use of a more relaxed threshold for interaction terms is not uncommon in the epidemiological literature. Importantly, as with all other analyses, our interpretation does not rely exclusively on p-values; rather, we report and discuss the estimated effects and confidence intervals.

In response to the reviewer's concern, we have now explicitly acknowledged in the Methods the reason for this choice and in the revised Discussion that the reliance on a significance threshold for interaction tests, and particularly the use of $p \leq 0.10$, is a limitation.

Methods

A P-value of ≤ 0.05 was considered statistically significant for all estimates, except for the interaction between the first-group exposure and number of previous events when we considered a $P \leq 0.10$. We decided a priori to increase the type 1 error rate to 10% because of the expected low power to test for interaction in this scenario,^{32,33} particularly considering a 2×4 categorical interaction terms in the model.

Discussion

It is worth mentioning that we may have been underpowered for some analyses of the humoral response, particularly those examining the interaction between the first-exposure group and the number of previous exposures as well as when evaluating complete exposure history. Although we a priori set the type I error for the interaction test to 10% to increase sensitivity, simulation studies showed that this does not increase power to detect interactions,⁴⁶ thus these analyses should therefore be considered exploratory.